# Central Tethyan platform-top hypoxia during Oceanic Anoxic Event 1a

Alexander Hueter[1], Stefan Huck[2], Stéphane Bodin[3], Ulrich Heimhofer[2], Stefan Weyer[4], Klaus P. Jochum[5], and Adrian Immenhauser[1]

[1]Institute for Geology, Mineralogy und Geophysics, Sediment and Isotope Geology, Ruhr-University Bochum, Germany
[2]Institute for Geology, Leibniz University Hannover, Germany
[3]Department of Geoscience, Aarhus University, Denmark
[4]Institute for Mineralogy, Leibniz University Hannover, Germany
[5]Climate Geochemistry Department, Max Planck Institute for Chemistry, Mainz, Germany

*Correspondence to:* Alexander Hueter (alexander.hueter@rub.de)

**Abstract.** Short-term hypoxia in epeiric water masses is a common phenomenon of modern marine environments and causes mass mortality in coastal marine ecosystems. Here, we test the hypothesis that during the early Aptian, platform-top hypoxia temporarily established in some of the vast epeiric seas of the Central Tethys and caused, combined with other stressors, significant changes in reefal ecosystems. Potentially interesting target examples include time intervals characterized by the demise of lower Aptian rudist-coral communities and the establishment of microencruster facies as previously described from the Central and Southern Tethys and from the proto-North Atlantic domain. These considerations are relevant as previous work has predominantly focused on early Aptian basinal anoxia in the context of the Oceanic Anoxic Event (OAE) 1a, whereas the potential expansion of the oxygen minimum zone (OMZ) in coeval shallow water environments is underexplored. Well known patterns in the $\delta^{13}C$ record during OAE 1a allow for a sufficiently time-resolved correlation with previously studied locations and the assignment to chemostratigraphic segments. This paper presents and critically discusses the outcome of a multi-proxy study (e.g., REE, U isotopes and redox sensitive trace elements) applied to lower Aptian shallow water carbonates today exposed in the Kanfanar Quarry in Istria, Croatia. These rocks were deposited on an extensive, isolated high in the Central Tethys surrounded by hemi-pelagic basins. Remarkably, during chemostratigraphic segment C2, the depletion of redox sensitive trace elements As, V, Mo, and U in platform carbonates, deposited in normal marine oxic waters, record the first occurrence of basinal, organic-rich sediment deposition in which these elements are enriched. During the C3 segment, seawater oxygen depletion established on the platform top as indicated by the patterns in Ce/Ce* and U isotopes. Shifts in redox sensitive proxies coincide with the expansion of microencruster facies. Segment C4 witnesses the return to normal marine reefal faunas on the platform top and is characterized by patterns in redox sensitive proxies typical of normal marine dissolved oxygen levels. It remains unclear, however, if platform-top hypoxia resulted from the expansion and upwelling of basinal, oxygen-depleted water masses, or if spatially isolated, shallow hypoxic water bodies formed on the platform. Data shown here are relevant as they shed light on the driving mechanisms that control poorly understood faunal patterns during OAE 1a in the neritic realm and provide evidence on the intricate relation between basinal and platform-top water masses.

**1 Introduction**

About 500 oxygen-depleted coastal zones have been described from the modern glaciated world (Altieri and Diaz, 2019). Of these, 180 are characterized by seasonal stratification, 62 show sub-annual events, and 61 recorded short-term episodic oxygen depletion (Diaz and Rosenberg, 1995; Diaz et al., 2011). This type of shallow seawater dissolved oxygen depletion must be clearly separated from recent anoxic settings such as the Black Sea, characterized by permanent basinal anoxia but overlain by about 100 m of oxic surficial waters (Zenkevich, 1963). The abundance of modern oxygen-depleted coastal water masses implies that the threshold limits from oxic to hypoxic seawater conditions are easily reached when at least one of the following environmental parameters is present: (i) shallow stratified waters, (ii) high nutrient availability, and/or (iii) elevated seawater temperature (Breitburg et al., 2018).

Intrigued by the abundance of recent anoxic shallow shelf water bodies, this study raises three fundamental questions: (i) Was shallow water hypoxia an equally (or even more) significant feature of Mesozoic epeiric-neritic seas? (ii) If so, how would this feature affect carbonate-secreting, (sub)tropical reefal ecosystems? (iii) What archives or proxies have the highest potential to record these conditions? With regard to the time interval of interest, greenhouse periods of the Cretaceous world and particularly time intervals with global oceanic anoxic events (Jenkyns, 1980) are perhaps among the most obvious targets to start with. During time intervals characterized by high eustatic sea level, globally elevated temperatures, and demonstrated restriction of widespread epeiric-neritic shelf water masses from open oceanic `blue waters´ (see Immenhauser et al., 2008 and references therein), the threshold limits that characterize shallow oxygen depleted water bodies of the recent world (Wu, 2002) are of great interest.

Here the focus is on the early Aptian of the Central Tethyan realm. Perturbations of the global carbon cycle spanning the middle part of the early Aptian culminated in the Oceanic Anoxic Event (OAE) 1a (120 Ma, with a duration of ~1.0 to 1.3 Myr; Li et al., 2008) and affected facies development on carbonate platforms across the world (Wissler et al., 2003; Burla et al., 2008). Rudist-coral platform ecosystems declined along most of the northern Tethyan margin (Föllmi et al., 2006) and probably also in the New World, whereas *Lithocodium aggregatum* and *Bacinella irregularis* or similar, taxonomically poorly constrained microencruster facies (Schlagintweit et al., 2010 and references therein), expanded in many lower palaeolatitude areas (Huck et al., 2010; 2011). Moreover, a coeval biocalcification crisis, caused by oceanic anoxia, affected pelagic organisms (Weissert and Erba 2004; Burla et al., 2008) and culminated in the worldwide deposition of organic-rich sediments (black shales). Much of the previous work, however, has focused on basinal anoxia (Schlanger and Jenkyns, 1976; Weissert and Erba, 2004).

This study utilized shallow water sections of the central Adriatic Carbonate Platform in NW Croatia (Kanfanar section, Istria; Huck et al., 2010). There, evidence for out-of-balance reefal ecosystems (ecosystems that recorded a sudden change in dominant biota) is found in the form of a demise of coral-rudist facies and a transient expansion of the microencrusters *L. aggregatum* and *B. irregularis*. Remarkably, this event is coinciding with the C3 chemostratigraphic segment (*sensu* Menegatti et al., 1998) as based on $\delta^{13}C_{carb}$ data. The C3 segment is characterized by a pronounced negative $\delta^{13}C$ excursion and marks the onset of OAE 1a (Huck et al., 2010). Excellent outcrop conditions along quarry walls allow for an in-depth documentation, description, and interpretation of platform-top carbonate facies across OAE 1a. The question raised is if time intervals of depleted

platform-top seawater oxygen levels periodically exceeded the tolerance levels of even the hardiest among Aptian coral and rudist reef builders? If so, was oxygen-depletion a key factor triggering the mass occurrences of microencruster organisms? This is relevant, as up to now, the factors that caused lower Aptian rudist-coral reefal ecosystem shutdown (Immenhauser et al., 2005; Rameil et al., 2010) in the Central and Southern Tethys and the proto-Atlantic domain (Huck et al., 2012) and the coeval expansion of microencrusters remain unknown.

To test these hypotheses, we apply palaeoecological, sedimentological, and geochemical proxies to fine-grained platform carbonates. These include Rare Earth Element (REE) concentrations and specifically the evolution of the Ce anomaly. This proxy has been applied to modern and ancient carbonates, and particularly microbialites, and has been successfully used to interpret the redox state of marine and lacustrine waters (Olivier and Boyet, 2006). Furthermore, we utilize U isotope ratios (Weyer et al., 2008) as a paleoredox proxy (Montoya-Pino et al., 2010). This tool bears evidence on the seawater signature of biogenic carbonates, and most importantly fractionation under hypoxia/anoxia (Romaniello et al., 2013). Moreover, redox sensitive trace elemental patterns (Bodin et al., 2007) were used and applied to selected carbonate materials from the Kanfanar quarry. Data shown here are relevant for those interested in Cretaceous anoxia in general and specifically significant for the understanding of the interaction of palaeoenvironmental processes in coastal and basinal settings.

## 2 Geotectonic and stratigraphic setting

### 2.1 Geotectonic setting

The 'Adriatic microplate' (Fig. 1A) formed as a result of Middle Triassic rifting (Dercourt et al., 1986) and initiated the formation of the Adriatic Carbonate Platform, one of the largest Mesozoic platforms of the peri-Mediterranean region, built by up to 8000 meters of sedimentary rocks (middle Permian to Eocene; Korbar, 2009). The Adriatic microplate drifted eastwards until the Late Cretaceous (Husinec and Sokač, 2006). The collision with Eurasia resulted in the uplift of the peri-Adriatic mountain chains. Compared to other parts of the Adriatic Carbonate Platform, Istria remained tectonically relatively stable (Korbar, 2009). The carbonate rocks studied here form part of the WNW domain of the Adriatic Carbonate Platform. Based on palaeogeographic reconstructions, shallow water sediments were deposited on a very large, mainly WNW-ESE trending, isolated high (Fig. 1A) that was surrounded by hemi-pelagic basins on all sides.

### 2.2 Stratigraphic setting

The Istrian carbonate succession can be divided into four transgressive-regressive megasequences (Velić et al.,1995). Each of those sequences is terminated by a major phase of subaerial exposure, representing a hiatus of 11 to 19 Myr duration (Vlahović et al., 2003). The active quarry of Kanfanar provides access to an upper Lower Cretaceous carbonate succession that comprises parts of the Barremian Dvigrad Unit as well as the lower Aptian Kanfanar Unit (Fig. 3). The Kanfanar section has been described in detail by Huck et al. (2010) and here, only the most relevant information is provided. Please refer to cited work for details.

The overall stratigraphic thickness of Barremian carbonates in western Istria is about 75 meters of which the uppermost eight meters, the `Dvigrad Unit´, are exposed in the Kanfanar Quarry (Fig. 1B). The Dvigrad Unit consists of laminated fine-grained tidalites and peloidal wackestones/grainstones with rare biota. Above, the high-energy to lagoonal facies of the lower Aptian Kanfanar Unit is composed of high-energy coral and rudist

floatstones with intercalated peloidal grainstones. Further upsection, 14 meters of rhythmically bedded mudstones/wackestones and oncoidal *L. aggregatum/B. irregularis* (microencruster) floatstones with abundant orbitolinid foraminifera follow. The microencruster-dominated strata are overlain by two meters of bioturbated wackestones/packstones with rare benthic foraminifera and echinoids. The stratigraphically uppermost four meters are only locally exposed and consist of bivalve floatstone facies (Fig. 3). The shoal-water deposits are overlain by a hiatus spanning the late Aptian to early Albian transition. The stratigraphy of the Kanfanar section is twofold and based on biostratigraphy ($\delta^{13}$C) of benthic foraminifera (Husinec et al., 2000; Vlahović et al., 2003; Velić, 2007) and dasycladacean algae (Velić; 2007), as well as chemostratigraphic data derived from $^{87}$Sr/$^{86}$Sr isotope ratios (Huck et al., 2010). Both approaches suggest late Barremian to early Aptian age for the upper part of the Dvigrad Unit and Barremian-Aptian transition age for the overlying Kanfanar Unit.

## 3 Methods

### 3.1 Fieldwork

We make use of previous studies providing detailed sedimentological and stratigraphical information (Huck et al., 2010). In order to complement existing data, a total of 41 carbonate rock samples from the Kanfanar section in Croatia were assessed in terms of their lithological composition, fossil content, and suitability for geochemical analyzes. In particular, we sampled automicrite (Neuweiler and Reitner, 1992), i.e. micrite that formed *in situ* by microbial activity inducing the nucleation and precipitation of mainly fine-grained calcite crystals. This approach was motivated by the fact that the geochemical proxies applied in the lab (REE, U isotope ratios and redox sensitive trace elements) have been documented to be most successful when applied to this type of carbonate material (Della Porta et al., 2015; Guido et al., 2016). Automicrite is differentiated from detrital micrite, i.e. fine-grained carbonate ooze that accumulated due to gravitational settling from the water column (Turpin et al., 2012; 2014). Mesoscopic differentiation of automicrite from detrital micrite in the field include a layered structure and a uniform mineralogical composition for automicrite, in contrast to fine grained detrital micrite characterized by mineralogically different constituents and variable grain size.

### 3.2 Optical Methods

Thin sections were analyzed in detail for the presence of *L. aggregatum* and *B. irregularis* and their corresponding growth forms. Please refer to Schlagintweit et al. (2010) for a discussion on the taxonomy of these organisms. In the sense of an umbrella term, we refer to both organisms as `microencruster´ facies throughout this paper. Automicrite samples as specified in the field were critically screened and only micrite of a fine-grained, homogeneous nature was selected for further analysis. Microfacies analysis was based on component analysis with the objective to separate stratigraphic intervals characterized by normal marine composition of biota (corals, rudists, echinoderms and gastropods) from intervals lacking normal marine biota but yielding very abundant microencrusters. Biogenic components in thin sections studied were assessed for their frequency distribution in a semi-quantitative manner using the following terminology: very abundant, abundant, rare, and absent to describe the relative volumetric significance of different components. The limestone classification of Dunham (1962) was applied.

Scanning electron microscope (SEM) analysis was used to examine crystal structure and sorting of crystal sizes. Six samples were examined using a high-resolution field emission scanning electron microscope (HR-FESEM type LEO/ZEISS 1530 Gemini) with an extremely high electrical voltage (EHT) of 20 kV at the Ruhr-University Bochum. The samples were sawn into small blocks, glued to slides and coated with gold. A conductive paste of silver was applied to prevent charges. The freshly cracked top was then examined for the presence of automicrite and detrital micrite.

**3.3 Geochemical Methods**

LA-ICP-MS analysis were performed on 33 samples with a 213 nm, Q-switched, Nd:YAG laser from New Wave, connected to a Thermo Finnigan ELEMENT2 sector field (SF) ICP-MS located at the Max Planck Institute for Chemistry in Mainz, Germany. The main focus of the analysis was on obtaining data for calculation of the cerium anomaly (Ce/Ce*). This ratio is used as a proxy for the degree of seawater oxygenation. Measured isotopes include Rare Earth Elements and redox sensitive trace elements. Every sample was measured at 3 spots in 2 areas of the thick section. Standard settings of the laser system were: (1) an energy density of 7 J/cm$^2$; (2) a pulse rate of 10 Hz; (3) low oxide production rates (ThO/Th < 0.5 %); (4) laser spot diameter size of 60 – 120 µm; and (5) wash out, blank count rate and ablation times of 45 s, 20 s and 110 s, respectively. We used NIST-612 and MACS-3 as our calibration material and $^{43}$Ca as an internal standard, to calculate absolute element concentrations from signal intensities. REE abundances were normalized to the Post Archean Australian Shale standard (PAAS) values given in Barth et al. (2000).

Ce anomaly (Ce/Ce*) was defined following Nozaki's (2008) calculation:

(1) Ce/Ce* = 2Ce$_N$ / (La$_N$ + Pr$_N$)

where "N" stands for shale-normalized concentration.

All 16 samples for U isotope analysis were powdered and weighted before digestion. For the carbonates about 600 mg up to 1 g were digested with 6 M HCL in 15 ml Salivex® beakers on a hot plate at 100 °C and dried overnight. Afterwards, samples were treated with 10 ml of 1.5 M HNO$_3$ and centrifuged after 15 minutes at ultrasonic bath. The residue was quantitatively transferred into 90 ml Salivex® beakers, treated with a mixture of 800 µl conc. HNO$_3$ and 3 ml conc. HCL (*aqua regia*), boiled up at 120 °C for 2 hours and evaporated afterwards. All samples were dissolved in 2 - 5 ml of 3 M HNO$_3$. Since the U concentration was known from LA-ICP-MS, all samples were spiked prior to chemical separation of U from the matrix with a $^{236}$U/$^{233}$U mixed isotopic tracer in order to correct for any isotope fractionation on the column and instrumental mass bias of the MC-ICP-MS. See Weyer et al. (2008) and Romaniello et al. (2013) for details on the spiking protocol.

Uranium separation from the host sample matrix was performed by chromatographic extraction, using Eichrom UTEVA resin modified after Horwitz et al. (1993). For further information about the chemical separation of U from the samples matrix, the reader is referred to Weyer et al. (2008).

Uranium isotope analyses were performed with a ThermoScientific Neptune MC-ICP Mass Spectrometer at the Institute of Mineralogy at Leibniz University Hannover, Germany. Analyses were performed using an Aridus II combined with a 50 or 100 µl ESI nebulizer for sample introduction. With this setup a 50 ppb U solution was

sufficient to achieve a 36 V signal on $^{238}$U. All U isotope variations are reported relative to the U isotope composition of the CRM112a standard. The results of the isotope measurements are provided in the delta-notation:

(2) $\delta^{238}U$ in ‰ = [ $(^{238}U / ^{235}U)_{sample} / (^{238}U / ^{235}U)_{standard}$ -1 ] x 1000

For detailed information on the U isotope analysis, the reader is referred to Weyer et al. (2008) and Noordmann et al. (2015).

## 4 Results

### 4.1 Automicrite *versus* detrital micrite

The term "automicrite" was defined by Neuweiler and Reitner (1992) and refers to micrite that formed *in situ* by microbial activity inducing the nucleation and precipitation of mainly fine-grained Mg calcite crystals. Both types of micrite can be found in various abundances throughout the Dvigrad and Kanfanar units. The volumetrical significance of automicrite increases markedly in the stratigraphic intervals characterized by microencruster facies (specifically the C3 chemostratigraphic interval of OAE 1a *sensu* Menegatti et al., 1998; Fig. 3). Under SEM (Fig. 2A, C), automicrite is typified by its homogenous grain size and the lack of skeletal debris. In contrast, detrital micrite is characterized by poor sorting of crystals and the presence of skeletal grain fragments (Fig. 2B, D). Detrital micrite is abundant in the Dvigrad Unit and the upper part of the Kanfanar Unit, both characterized by tidal flat, high-energy/rudist facies (Fig. 3). See also Turpin et al. (2012; 2014) for further criteria for the interpretation of fine-grained carbonates.

### 4.2 Microfacies analysis

The lower part (Dvigrad Unit; 8 to 14 m) includes inter- to supratidal, weakly laminated, peloidal packstones/grainstones as well as rudist- and coral floatstones. This high-energy facies is dominated by benthic foraminifers, corals, rudists, gastropods and echinoderms (Fig. 3).

Microencruster facies is present in section meters 14 to 26 (Kanfanar Unit; Fig. 3). This part is characterized by a subtidal, low-energy setting with short interruptions made up of lower subtidal, protected lagoon, low-energy facies with increased sedimentation rate, as seen by a change to more vertical microencruster growth forms. Other carbonate secreting marine organisms (rudists, corals, echinoderms, foraminifera etc.) are remarkably scarce in these intervals. The upper part of the Kanfanar Unit (26 to 32 m) is composed of high-energy grainstones/rudstones with intraclasts and reworked material as well as rudist- and coral floatstones. The biota found here largely resemble those found in the Dvigrad Unit (Fig. 3).

### 4.3 Redox sensitive trace elements

Analyzed redox sensitive trace elements include arsenic (As), vanadium (V), molybdenum (Mo), and uranium (U). Following the protocol of previous authors (Olivier and Boyet, 2006; Collin et al., 2015), we targeted automicrite as the archive of choice. The disadvantage of this approach is that we are limited to a chemostratigraphic section

that corresponds to section meters 7 to 31, in other words, the interval that yields volumetrically significant portions of automicrite. In the lower part of the section (8 to 14 m), the above mentioned elements reveal an increase in their concentration. The concentration of As increases from 0.7 (8.2 m) to 3.2 ppm (13.2 m), that of V from 8.0 (8.2 m) to 17.6 ppm (13.2 m) and that of Mo from 3.5 (8.2 m) to 7.2 ppm (12.8 m; Fig. 4). This trend is followed by a decrease to concentrations similar to that found at section meter 8.2 and coincides remarkably well with the onset of the microencruster facies (section meter 14, Fig. 4). Concentrations of those elements remain low throughout the C3 segment, arguably showing an increase in the uppermost sample. The concentration of U is decreasing from 1.98 (8.2 m) to 0.44 ppm (27.9 m) and, similar to the other redox sensitive elements, shows an increase in the uppermost sample (Fig. 4).

### 4.4 Rare Earth Elements (cerium anomaly)

The lowest stratigraphic interval (section meters 8 to 14) is characterized by Ce anomalies with values between 0.4 and 0.6 (Fig. 5). The onset of the microencruster interval is characterized by similar values. In the upper portion of the section (section meters 20 to 26), the Ce anomaly values shift progressively from 0.4 to 0.95, reaching maximum values in the uppermost part of the microencruster interval at 25.5 m. The top of the Kanfanar Unit (section meters 26 to 28) is characterized by a return of Ce anomaly values to pre-excursion values of 0.4. In an attempt to test whether the Ce stratigraphic pattern represents a significant feature, the lanthanum (La) anomaly was calculated and results are depicted in figure 6. The aim is to test whether the Ce anomaly values are genuine or an artifact caused by elevated amounts of La (Bau and Dulski, 1996; Bodin et al., 2013). The outcome is that the amplitude of the Ce anomaly values are moderately affected by elevated amounts of La. The data, however, also established that normal marine biota are stronger affected than microencruster facies. Please note that the La anomaly diagram was slightly modified by Webb and Kamber (2000). However, as this modification extends the boundaries for genuine Ce anomalies, we here apply the somewhat more critical approach by Bau and Dulski (1996).

### 4.5 Uranium isotope ratios

In the upper portions of the Dvigrad Unit, $^{238}U/^{235}U$ ratios increase from 0.25 to 0.46 ‰ (Fig. 5). Within the Kanfanar Unit, $^{238}U/^{235}U$ ratios start to decrease from 0.45 ‰ (at 14 m) to 0 ‰ (at 26 m). The onset of the OAE 1a-equivalent stratigraphic interval is characterized by a continuous decrease of $^{238}U/^{235}U$ ratios to values of 0 ‰. Above section meter 26, the Kanfanar Unit is characterized by normal marine biota, and an increase of U isotope ratios to values of 0.2 ‰.

### 5 Interpretation and Discussion

### 5.1 Stratigraphic framework

The biostratigraphy of the Kanfanar section and the possible connection between OAE 1a and microencruster occurrences are fundamental for the understanding of the very complex cause and effect patterns. Here, the

stratigraphy of the Kanfanar section is based on biostratigraphy (benthic foraminifera and dasycladacean algae) and chemostratigraphy ($^{87}Sr/^{86}Sr$). The $\delta^{13}C$ pattern (based on rudist shells of the Kanfanar section; Huck et al., 2010) is even more complicated by the absence of both, the base of the Aptian positive isotopic excursion, and the following prominent short negative spike. Please note that we have shifted the C3 and C4 segments relative to the Huck et al. (2010) paper. This interpretation is based on the first occurrence of microencrusting organisms on the platform domain, being coeval with the onset of hypoxic conditions in basinal settings expressed as organic-rich deposits comparable to black shales. The C3 segment is characterized by a pronounced negative $\delta^{13}C$ excursion and marks the onset of OAE 1a equivalent in Kanfanar. When comparing this pattern to the Cismon section (Menegatti et al., 1998) defined by the first occurrence of black shales, evidence for the assignment of the C3 segment in Kanfanar to the base of the Kanfanar Unit is found. The top of C3 as well as the base of C4 in Kanfanar are picked on top of the last microencruster occurrence based on the lowest seawater oxygen content, as suggested in Ce/Ce* and by means of redox sensitive trace element concentrations, as well as the $\delta^{13}C$ signatures of rudist low-Mg calcite as shown in Huck et al. (2010). Nevertheless, we are aware of the intriguing difficulties that related to the stratigraphy of shallow-water carbonate rocks. Along these lines of reasoning, Huck et al. (2010) compared the $\delta^{13}C$ pattern of the Kanfanar section with patterns of a composite section in Oman and the La Bedoule section in France, presenting many similarities to the Kanfanar section. The similarities in these patterns include the strong positive peak in $\delta^{13}C$ of +4 ‰ (rudist shells in the Kanfanar section) and support the here proposed interpretation of the stratigraphic framework, that is further supported by chemostratigraphic data based on $^{87}Sr/^{86}Sr$ ratios (Oman and Croatia).

Concluding, the connection of hypoxia (or even anoxia) and the rock-forming dominance of microencruster facies would not be affected by the possible interpretation of a pre-OAE 1a age. Note, *L. aggregatum* and *B. irregularis* have been described prior to and after OAE 1a (e.g. in Portugal and Oman). Low oxygen concentrations, independent of the oceanic anoxic event 1a, remain the, in the view of the authors, most likely cause for the ecological patterns observed.

**5.2 Redox proxies applied to platform-top automicrite**

For the reconstruction of palaeo-redox conditions, it is crucial to ensure a marine hydrogenous source and to avoid a detrital origin. In order to ensure that this is the case, all samples were carefully investigated (by means of thin sections and SEM analysis) for the presence of automicrite. We here follow previous workers who have clearly documented that automicrite is an outstanding archive of paleo-seawater geochemistry, notably for its REE pattern (Olivier and Boyet, 2006; Della Porta et al., 2015). Recrystallization or aggradational neomorphism, leading to a shift of values away from marine patterns, is recognized by means of an inhomogeneous grain size distribution. Evidence for this feature is lacking in the samples studied here.

The majority of REE patterns measured in automicrite in the context of our study reveal a typical marine pattern (Fig. 7), supporting the interpretation of a hydrogenous (marine) source. This interpretation is supported by relatively high Y/Ho ratios with values between 40 and 50, indicative of a marine source. Moreover, the ratio of Light Rare Earth Elements (LREEs) and Heavy Rare Earth Elements (HREEs) is below 1.0 for all measured samples, a feature that is once more typical for a marine signal (Fig. 7). In general, the Al content (Fig. 4) is low and oscillates at values around 500 ppm (0.05 ‰), significantly below the suggested maximum for a hydrogenous source of 0.35 ‰ (Ling et al., 2013). The same applies for Sc. The highest measured concentration in our samples

is close to 0.3 ppm, much lower than the maximum concentration (for seawater) of 2 ppm, as suggested by Ling et al. (2013).

For the interpretation of the redox proxies, it is crucial to understand that these must be viewed at different temporal levels. Uranium isotope ratios will preserve a global signal due to the long residence time of U in the global ocean (400 kyr; Ku et al., 1977). In contrast, Ce, with a relatively short residence time of a few hundred years, as well as the redox sensitive trace elements, will display a local or even regional signal (Sholkovitz and Schneider, 1991; Shields and Stille, 2001; Bodin et al., 2007), supported by the long mixing time of the oceans (~ 1000-2000 years).

Increasing concentrations of redox sensitive trace elements in the C2 chemostratigraphic segment (Dvigrad Unit; Fig. 4) are tentatively interpreted as harbinger of increasingly oxygen-depleted water masses. Increased concentrations of Mo, V, As, and U are commonly assigned to anoxic environments because in the reduced state, the solubility of V, Mo, and U decreases, and their enrichment in sediments is favored (Algeo and Maynard, 2004; Bodin et al., 2007; see also Huerta-Diaz and Morse (1992) for a mechanism of As uptake in anoxic environments). In the case example discussed here, the redox sensitive trace elements, As, V, Mo, and U indirectly show the redox conditions on the platform top. In increased uptake of these elements during basinal black shale deposition, will reduce their concentration in the seawater reservoir of the platform top, as seen in the rapid decrease of these elements in chemostratigraphic segment C3 at the onset of OAE 1a (cf. Algeo, 2004). The onset of the C4-equivalent segment in Istria is characterized by a moderate increase in the concentration of redox sensitive trace elements, representing the transition from microencruster facies to normal marine, here arguably more oxygenated, deposits. Due to the scarcity of automicrite in the upper portions of the C3 segment, the partially incomplete trace element pattern is supported by independent thin section evidence (biota; Fig. 3).

In addition to trace elements, the understanding of the REE distribution in seawater through time has allowed to constrain relationships and interactions of the atmospheric-lithospheric-oceanic system (Kamber, 2010). The behavior of Ce can be used to interpret palaeo-seawater oxygenation levels (Bodin et al., 2013). In modern-day oxygenated shallow open seawater, the Ce/Ce* ratio is close to 0.4 (Nozaki, 2008). Values approaching 1 indicate oxygen depleted conditions. During the Hauterivian-Barremian, Ce/Ce* values of open ocean seawater have arguably fluctuated between 0.3 and 0.6, mostly as a function of primary productivity (Bodin et al., 2013). In the early Aptian, values approaching 0.8 are observed coeval to the unfolding of OAE 1a in the open ocean. In Croatia, at the onset of segment C3, values around 0.4 are found in samples characteristic of a low-energy lagoonal facies, and coincide with first occurrences of *L. aggregatum* and *B. irregularis* oncoids, as well as orbitolinids and echinoids. Whereas this value points to normal marine oxygenated conditions, the subsequent increase to a value close to 1 (see Nozaki, 2008), starting at meter 20, arguably reflects a decrease in seawater dissolved oxygen content. On the level of a tentative working hypothesis, the maximum value of 0.97 at meter 25.5 may coincide with the transition from the C3 to the C4 chemostratigraphic segment. This transition is marked by the onset of a positive $\delta^{13}C$ excursion, induced by increased organic carbon burial (Menegatti et al., 1998), arguably reflected in our data. At the top of the C3 segment, U isotope ratios and Ce anomalies indicate a maximum oxygen depletion, but the volumetrical significance of microencrusting facies is decreasing. The two possible explanations are: (1) The complex morphology of *L. aggregatum* and *B. irregularis* (e.g. Rameil et al., 2010) may induce some degree of data bias in thin sections, i.e. the thin section area taken up by microencrusters strongly depends on the plane of observation. Moreover, their abundance and their growth morphology may change due to environmental patterns. The authors are aware of this problem and took great care to compare observations from the field and thin sections to limit the error bar to the possible minimum. Despite of this very conservative approach, we cannot

exclude that some of the second order patterns in microencruster abundance are the result of a data bias. We are, however, confident that the first order patterns are valid and can be replicated. (2) The second and more likely interpretation is: Seawater oxygen depletion is reaching its maximum at the top of C3 and becomes critical for even the hardiest organisms (here microencruster facies), exceeding their tolerance limits. Hence, peak values of anoxia may coincide with a decrease, rather than an increase in microencruster abundance. The decrease of the microencruster facies and the return of normal marine biota is characterized by a decreasing Ce anomaly to values that are commonly assigned to oxygenated seawater values. The Ce anomaly was compared to the Pr anomaly to verify whether negative Ce anomaly values are created artificially by La enrichment or genuine by Ce depletion (Bau and Dulski, 1996; Shields and Stille, 2001; Fig. 6). The outcome of this test is that La enrichment is present in more than 50 % of the samples and where so, negative Ce anomalies might be slightly overestimated. Another important outcome is that samples dominated by normal marine biota are seemingly more affected by positive La anomalies (Fig. 6, field IIa) compared to samples dominated by microencrusters.

The Ce/Ce* interpretations are confirmed by the trend of U isotopes in the Croatia section. Indeed, dissolved U in seawater is a conservative ion with a residence time of ca. 400 kyr. (Ku et al., 1977). Uranium exists in two redox states: (i) soluble U (VI) found in aqueous complexes with calcium and carbonate ions in seawater and (ii) insoluble U (IV) which is reduced and removed within the sediment. Differences in $^{238}$U and $^{235}$U arise during the reduction of U (VI) to U (IV), where $^{238}$U is preferentially removed and reduced from solution due to a nuclear volume effect. Uranium isotopes do not experience significant mass-dependent fractionation. According to its redox-sensitive character, the abundance and isotope composition of U recorded in sediments are used as proxies to reconstruct seawater redox patterns (Weyer et al., 2008; Jenkyns, 2010; Andersen et al. 2014). Fluctuations in the relative importance of the individual sinks driven, for example, by the enhanced occurrence of seafloor anoxia, should be recorded in the U isotope composition of the individual sinks.

Relative to modern open marine seawater ($\delta^{238}$U of -0.41 ± 0.03 ‰; Weyer et al., 2008), the $^{238}$U/$^{235}$U ratios of the Kanfanar section are elevated. A possible explanation might be an isotope fractionation in the context of U reduction. Uranium isotope variations in ancient carbonates are perhaps controlled by changes in seawater pH, $PCO_2$, $Ca^{2+}$, or $Mg^{2+}$ concentrations (Chen et al., 2016). Hood et al. (2018) showed that different carbonate particles display a high variability in $^{238}$U values, even within a single sample. Some modern bulk sediments are slightly offset from seawater (+0.4‰), attributed to minor incorporation of U (IV) into the carbonate lattice (Chen et al., 2018). Microbial reduction of U (VI) to U (IV) under anoxic conditions at the sediment-water interface results in a decrease in U solubility and a change in $^{238}$U/$^{235}$U. The U isotopy will fractionate to heavier values during incorporation of U into the carbonate under hypoxic/anoxic conditions in the upper water column. In case of an oxic upper water column, the anoxic zone within the carbonates would be rather deep, leading to the absence of a significant diagenetic U mobilization (Romaniello et al., 2013; Chen et al., 2018). These processes or settings, in isolation or combination, might explain an increase in the $^{238}$U/$^{235}$U ratio from 0.3 up to 0.7 ‰ (Stylo et al., 2015). Another possible factor influencing the $\delta^{238}$U would be a shuttle of fractionated U from an anoxic seawater mass to the carbonate platform aquafacies. An explanation for this shuttle could be a climatically induced change in circulation patterns. Martin et al. (2012) demonstrated a shift in Nd isotopes at the Demerara Rise during the Late Cretaceous greenhouse period, contemporaneous to OAE 2. This shift was arguably triggered by a change in circulation patterns, increasing downwelling in the Tethys and upwelling in the Demerara region. This overall setup possibly acted as a dynamic nutrient trap, contributing to the development of organic-rich sediments and thus, hypoxic/anoxic conditions. The $\delta^{238}$U value of seawater U (VI) decreases as the spatial extent of bottom

water anoxia increases (Montoya-Pino et al., 2010). Generally, $^{238}U/^{235}U$ ratios in our data display a similar pattern as that observed for the Ce anomaly (Fig. 5). Decreasing ratios throughout the C3 segment are followed by an increase in the section meters 26 to 28, characterized by the decline of the microencruster facies. Similar to the above proxies, this is indicative of a return to normal marine oxygen isotope levels.

Concluding, the multi-proxy approach shown here includes both biotic and geochemical tools. All of these proxies agree with a decrease in platform-top water dissolved oxygen levels with a peak oxygen deficiency in the C3 chemostratigraphic segment (25.5 m) coinciding with the lower Kanfanar Unit. Having established redox patterns in lower Aptian platform-top settings in Istria, data shown here must be placed into a mechanistic context.

## 5.3 Principles of recent coastal hypoxia and tentative applicability to Aptian neritic platform water masses

In this paper, we follow the definitions of Naqvi et al. (2010 and references therein) with regard to seawater dissolved oxygen levels: oxic (8-1.4 ml/L $O_2$), hypoxic (1.4-0.1 ml/L), suboxic (< 0.1 ml/L), and anoxic (0.0 ml/L). Along recent coasts, hypoxia defines periods when seawater dissolved oxygen levels fall below 1.4 ml of $O_2$/liter. Often, this takes place during summer months (both in the northern and southern hemisphere, respectively) when air and seawater temperatures reach maxima. In contrast, during colder months, normal dissolved seawater oxygen concentrations are between 4 and 6 ml of $O_2$/liter (Rabalais et al., 2001). At a value below 2 ml of $O_2$/liter, i.e. within the lower oxic level, benthic faunas start to display aberrant behavior (Alvisi et al., 2013 and references therein). An example from the Venezuelan coast shows that 60 to 98 % of corals were annihilated during a short term (several days) period of hypoxia (Laboy-Nieves et al., 2001), whereas macroalgae showed little signs of degradation. Whereas some marine organisms are more resistant to short intervals of moderate or even severe hypoxic conditions (see a summary in Diaz and Rosenberg, 1995, Table 2), the demise of nearly all normal marine biota (Fig. 2), such as described from the lower Aptian sections in Istria, indicates a fundamental, long-term environmental pattern.

In this context, it is relevant that the taxonomically difficult microencruster facies, replacing coral-rudist assemblages, is presently interpreted as microbial consortium (Schlagintweit et al., 2010). Lacking a modern analogue to *L. aggregatum* and *B. irregularis*, we refer to studies dealing with the oxygen tolerance of modern bacterial communities. Examples include Wenger (2000) describing molecular sensors for oxygen in bacteria. Marine metazoans, in an attempt to maintain oxygen delivery, have several strategies to respond to hypoxia. Attempts are made to reduce energy consumption and increase efficiency of key metabolic processes (Hochachka, 1997 and references therein). These physiological and biochemical adaptations arguably result in reduced growth rates, as demonstrated for echinoderms (*Amphiura filiformis*) and bivalves (*Crassostrea virginica* and *Mytilus edulis*; Diaz and Rosenberg, 1995). This in turn strongly reduces the metabolic activity of these organisms and affects chances of survival whereas specifically adapted microbial communities can thrive and outcompete the weakened metazoans. We emphasize that it is not possible to clearly separate the effects of hypoxia from related environmental stressors (Wu, 2002) that may or may not have affected Aptian reefal ecosystems. These include changes in trophic levels, very high seawater temperatures, and changes in salinity, all representing stressors for most reefal organisms. In present day shallow shelf waters, dissolved oxygen depletion near the seafloor occurs as a result of seasonal water column stratification. The downward mixing of well-oxygenated surface water is limited with the result that dissolved oxygen near the bottom can become depleted due to biological consumption (Alvisi

et al., 2013). Thus, the establishment of hypoxia is also influenced by nutrient availability (Conley et al., 2009) including phytoplankton blooms (Druon et al., 2004) and, in the Recent, by human activity.

In the context discussed here, it is relevant that intermittent periods of seawater oxygen depletion in the Recent can occur on open shelves (e.g., the middle Atlantic Bight or the northern Gulf of Mexico). In modern settings, shelf-slope fronts seal off the shelf bottom water at its ocean-ward margin, and hence limits its circulation with oxygen-rich open water masses, so that the oxygen demand on the shallow shelf exceeds the ability of cross shelf circulation (Tyson and Pearson, 1991). Given that Aptian carbonates in Istria represent the deposits of a very large, isolated carbonate edifice with a WNW to ESE extension of ~1800 km's and a WE extension of perhaps 200 kilometers (Fig. 1), these considerations are of relevance here. Obviously, such a massive topographic feature represents a significant obstacle to ocean currents and tidal waves. Judging from recent, albeit much smaller isolated structures (seamounts etc.), tidal waves and wind-driven water circulation is significantly affected by topographic features of this type (Boehlert, 1988 and references therein). Moreover, the very significant dimensions of the neritic platform area ($> 500.000$ km$^2$) makes it likely that the Adriatic Carbonate Platform was topographically differentiated and segmented into hydrodynamically more restricted areas, separated by seaways acting as main passages for tidal currents. This feature can be clearly seen by comparing the outcome of this study with that of two previous ones (Husinec et al., 2012; 2018) from the southern extension of the same carbonate platform (Mljet section). There, a 10-m-thick interval was documented that arguably is in agreement with deposition under hypoxic conditions. The interval spans from the upper part of the C4 to the top of the C7 segment. The significant geographical distance (~1000 km) between the Kanfanar and Mljet sections, however, is in agreement with the concept of regional patterns riding on top of provincial or even global ones. In the view of the authors, a `global´ event is often the expression of the clustering of many regional (and even local) events that are more or less penecontemporaneous in nature in many basins. Moreover, in our view, the bulk $\delta^{13}$C signatures in the Mljet section have seen less diagenetic alteration compared to the section discussed here. Patterns as seen on the Adriatic Carbonate Platform are observed in many actualistic shallow marine settings worldwide (e.g., The Great Bahama Bank with an area of 103.000 km²). Bathymetric and facies maps of Bahama Bank point to areas dominated by tidal waves and currents alternating with domains that are hydrodynamically more isolated (Harris et al., 2015) and arguably only affected during major storm events. Acknowledging this level of spatial diversity represents a reasonable foundation for the interpretation of data presented here.

During the early Aptian OAE1a, a preconditioned oxygen deficiency of bottom water masses resulted in the deposition of layered organic-rich deposits (black shales; Bersezio et al., 2002; Burla et al., 2008) in basinal settings. The here proposed establishment of transient periods of oxygen-depleted water masses on the Adriatic Carbonate platform allows for two scenarios (Fig. 8): (i) A temporal expansion of the basinal OMZ into the shallow basinal bathymetric domain and the advection of these waters onto the Adriatic platform, possibly due to upwelling, driven by tidal pumping, wind-induced currents, or a combination of several factors. This effect could also be triggered by a flow of hypersaline water masses leaving the platform top and driving lateral advection or upwelling of open-ocean waters onto the platform top. Assuming typical paleo-temperature reconstructions of the lower Aptian world (Larson and Erba, 1999; Jenkyns, 2003), a greenhouse setting associated with strong evaporation on the platform-top is likely. Alternatively, (ii) the establishment of local, oxygen-deficient shallow water masses over portions of the Adriatic platform must be considered. These were physically separated from anoxic waters in the adjacent basins or seaways acting as main passages for tidal currents.

Another aspect that merits attention is the exchange of oxygen between hydrosphere and atmosphere. Under the early Aptian greenhouse climate, combined with strong evaporation rates, ongoing carbonate production (due to the microencruster mass occurrence), as well as the production and integration of $CO_2$ in the open ocean (Ontong Java; Jenkyns, 2003) the loss of oxygen to the atmosphere might have surpassed the adsorption of oxygen into the surface ocean. This scenario refers to the kettle-effect described by Skelton and Gili (2012). Finally, the modern world knows more than 500 oxygen depleted coasts (Altieri and Diaz, 2019), in the view of the authors a clear argument to support the concept of shallow water mass oxygen depletion. Atmosphere-hydrosphere oxygen exchange also depends on wave activity, as waves are major agents in actively transporting gases into the ocean surface water. In a protected, low-energy setting with very limited wave activity (due to wave-seafloor interaction) and a shallow wave base, the formation of oxygen-depleted surface water seems likely. Concluding, both scenarios would allow for a decline of benthic, oxygen-sensitive organisms and the mass occurrence of microencruster species being more tolerant to hypoxia. Hypoxic events with a duration of days to weeks, as typical for recent costal systems, are sufficient to cause mass mortality of many marine organisms and recovery times of benthic communities are in the order of several years (Wu, 2002). This implies that both, the establishment of long-term (upper portions of C3 segment; Fig. 3) hypoxia or alternatively, the repetition of numerous punctuated hypoxic and oxic events over extended periods will cause benthic ecosystem collapse as observed in the Kanfanar section in Istria.

Partially similar considerations were presented for the North American Late Pennsylvanian Mid-continent Sea (LPMS; Algeo et al., 2008). In their comprehensive study, Algeo and co-workers contrasted and compared three modern epicontinental seas (Hudson Bay, Baltic Sea, and Gulf of Carpentaria) with the LPMS. A critical boundary condition unique to the LPMS was the preconditioned, oxygen-deficient nature of the intermediate water mass that was laterally advected into the Mid-continent Sea. The consequence was a marked shallowing of the oxygen-minimum zone and water mass `aging´ (see Immenhauser et al., 2008 for a discussion of water mass aging). Differences to the Aptian case example discussed here include the absence of patterns that characterize the super-estuarine circulation model of Algeo et al. (2008), a feature that requires a hinterland providing riverine input. Judging from the palaeogeographic reconstructions of the Aptian Tethys, the Adriatic platform (Fig. 1) was isolated and lacked a connection to a hinterland. The reconstructed position of the section studied here combined with regional geological evidence points to an elevated area at the northern edge of the Adriatic Carbonate Platform, an overall shallow marine setting (<10 m), and this concept is also supported by the nature of the carbonate facies. Therefore, water depths of many 10's of meters seem unlikely for this part of the platform.

In this context, the work by Skelton and Gili (2012) is perhaps equally important because it deals specifically with the nature of Aptian platform-top water masses. Given that global temperatures peaked during the C3 chemostratigraphic segment (and then cooled), the patterns described here have significance when applied to the work of Skelton and Gili (2012). The term `kettle effect´ was coined in this context to describe the thermal $CO_2$ expulsion from extremely warm, restricted platform-top water masses despite high levels of atmospheric $CO_2$. This feature arguably limited a drop in seawater pH in the platform-top domain allowing for significant carbonate production as documented by microencruster facies expansion found in Istria and in other locations. Recent international research initiatives such as BIOACID, documented that most marine metazoans tolerate one stressor to some degree (see for example bivalves, Hahn et al., 2012) pending that sufficient food is available. But, if for example high temperatures plus low seawater pH act in parallel, these organisms reach their tolerance limits. In a fossil setting such as the one described here, the fundamental question which among perhaps several stressors was

most lethal must remain unresolved to some degree. Circumstantial evidence may add information though. High carbonate production rates, for example, are not in agreement with low seawater pH. The decline of oxygen sensitive organisms and a dominance of organisms that are (arguably) less sensitive to low oxygen levels is another factor that should not be neglected. Summing up, based on the data presented here, it is argued that seawater oxygen depletion was the main driver of the ecological patterns observed.

**5.4 Chronology of OAE 1a anoxia in Croatia: Basinal *versus* platform-top water masses**

Based on the data shown here, the $\delta^{13}$C signatures of well-preserved rudist calcite (Huck et al., 2010), and applying the Menegatti et al. (1998) chemostratigraphic scheme as a temporal framework as well as information on basinal anoxia from published work (Schlanger and Jenkyns, 1976; Weissert and Erba, 2004; Bodin et al., 2013), we propose the following succession of events (Fig. 9):

(i) The pre-OAE1a setup: The C2 segment (Dvigrad Unit; Fig. 3) represented normal marine platform-top deposition with $\delta^{13}C_{carb}$ values oscillating at 2-3 ‰. The redox sensitive trace elements (As, V, Mo, and U) show only small variation in their abundance implying normal oxygenated conditions during the early C2 segment equivalent. Microencruster facies was yet rare or absent. In the basin, dissolved seawater oxygen became increasingly depleted at the end of segment C2, expressed by the increase in redox sensitive trace element concentrations (Bodin et al., 2007) that is also recorded in the platform top environment.

(ii) The first record of oxygen depletion in platform-top environments: The lower part of the C3 segment arguably marks the beginning of the OAE 1a equivalent. It must be noted, however, that the pattern between seawater chemostratigraphy and precursor black shale facies is complex and first organic-rich deposits are present in the top of the C2 segment (Fig. 5 of Menegatti et al., 1998; Cismon Section). Barremian regional black shales are abundant (Weissert et al., 1979; Giorgioni et al., 2015) and possibly impacted the geochemistry of shallow water masses. In the section studied here, this unit is characterized by a high-energy rudist facies (Fig. 3). From section meter 15 onwards, microencruster facies dominates a low-energy environment. The $\delta^{13}C_{carb}$ ratios decrease and reach values of 0 to 1 ‰. On the platform top, dissolved oxygen levels slowly decrease (Fig. 5, Ce/Ce*). In basinal settings, the establishment of an oxygen depletion zone is recorded (Weissert et al., 1979). Redox sensitive trace elements in platform-top automicrite display a significant shift to very low values, best explained by increasing sequestration in deep-marine black shales. This implies that processes active in the basinal domain are recorded in the platform environment. Microencruster facies is rare to absent but then rapidly expands around section meter 16, some meters above the C2/C3 equivalent boundary. This points to a lag effect between basinal anoxia and its effects in the platform domain.

(iii) OAE 1a equivalent platform-top water oxygen depletion: The upper portions of the C3 segment arguably coincides, in terms of atmospheric $CO_2$ injection into the atmosphere (Méhay et al., 2009; Skelton and Gili, 2012), with the climax of OAE 1a. In the Kanfanar section, the upper portions of the C3 segment are similar to the lower ones inasmuch as microencruster facies is still dominant. The $\delta^{13}C_{carb}$ values reach a minimum of -1 ‰. The Ce anomaly as well as the U isotope ratios (Fig. 5) both point to significantly oxygen depleted platform-top water masses. Redox sensitive elements still display low concentrations, pointing to ongoing oxygen depletion in the basin.

(iv) Decline of microencruster facies and return to oxygenated platform-top water masses (C4 to (?) C6): These units are present in the uppermost section meters of the Kanfanar Unit, commencing at meter 26 (Fig. 3).

The facies record shifts from a low- to a high-energy setting, comparable to the one characterizing the C2 segment. Bulk matrix carbon isotope values show an increase to values oscillating around 0 ‰. In contrast, $\delta^{13}C$ of well-preserved rudist calcite record values of 3-4 ‰ (Huck et al., 2010). The microencruster facies disappears and rudist bivalves, gastropods, echinoderms and other marine biota return in large numbers as observed in outcrops and in thin sections. Cerium values and $^{238}U/^{235}U$ ratios suggest normal oxygenated seawater on the platform top. Redox sensitive trace elements show a moderate increase in concentrations, suggesting the transition to more oxygenated conditions in the basinal domain.

## 6 Conclusions

This paper documents that during the early Aptian of the Central Tethys, platform-top seawater hypoxia was a likely stressor causing the collapse of rudist-coral ecosystems and the coeval expansion of microencruster facies. In the Kanfanar Quarry, Croatia, OAE1a-equivalent platform top carbonates, formerly deposited on an extensive but isolated high in the central Tethys are exposed. Proxies applied to automicrite samples taken from the quarry walls include $^{238}U/^{235}U$ and Ce/Ce* ratios, and redox sensitive trace elemental concentrations. Patterns in redox-sensitive proxies and faunal changes are assigned to the chemostratigraphic segments C2 through C4 as based on $\delta^{13}C$ values. The following succession of events is found: During chemostratigraphic segment C2, redox-sensitive trace elements in oxygenated platform environments record the onset of basinal, organic-rich sediment deposition. During the C3 segment, basinal seawater oxygen depletion reaches an acme and hypoxia now also affects platform top water masses as indicated by patterns in Ce/Ce* and $^{238}U/^{235}U$ ratios. Shifts in redox-sensitive proxies coincide with the expansion of microencruster facies and the decline of rudists and corals. It remains unclear if oxygen-depleted basinal waters were upwelled onto the platform, or if spatially isolated, hypoxic water bodies formed in the platform domain. Lessons learnt from recent shallow hypoxic coasts arguably support the latter scenario. Segment C4 witnesses the stepwise return to normal marine reefal faunas in the platform domain and is characterized by patterns in redox sensitive proxies typical of normal marine dissolved oxygen levels. We emphasize that hypoxia is probably but one out of many stressors likely affecting the vast shallow neritic seas of the Aptian world. The data shown here are relevant as they document that seawater oxygen depletion during ocean anoxic events is not an exclusive feature of basinal settings.

*Data availability.* Data used for this study are available upon request to the corresponding author (alexander.hueter@rub.de).

*Author contributions.* A. H. performed field and laboratory work, compiled and interpreted the data, wrote the manuscript and drafted the figures. S. H. developed the stratigraphy and contributed to data interpretation. S. B. contributed to analysis of the data and data interpretation for Rare Earth Elements and redox sensitive trace elements. U. H. contributed to data interpretation and analysis of the data. S. W. contributed to data interpretation for uranium isotopes. K. P. J. contributed to data interpretation for Rare Earth Elements and Trace Elements. A. I.

initiated the study and contributed to the analysis of the data and their interpretation. The co-authors contributed to the writing of the manuscript. Correspondence and request for materials should be addressed to A. H.

*Competing interests.* The authors declare that they have no conflict of interest.

*Acknowledgements.* We thank Nadja Pierau, Yvonne Röbbert, Annika Neddermeyer, and Lena Steinmann (Institute of Geology, Leibniz University Hannover, Germany) for their advice in uranium geochemistry and support in the laboratory. Special thanks to Brigitte Stoll and Ulrike Weis (Max Planck Institute for Chemistry, Mainz, Germany) for preparing the LA-ICP-MS measurements. We also thank the handling editor Yannick Donnadieu, and Thomas Algeo, Helmut Weissert, and Anton Husinec for their constructive reviews and comments. This project was funded by the German Science Foundation (DFG, Project IM44/19-1 and HU2258/3-1). We acknowledge support by the DFG Open Access Publication Funds of the Ruhr-Universität Bochum.

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

**Figures:**

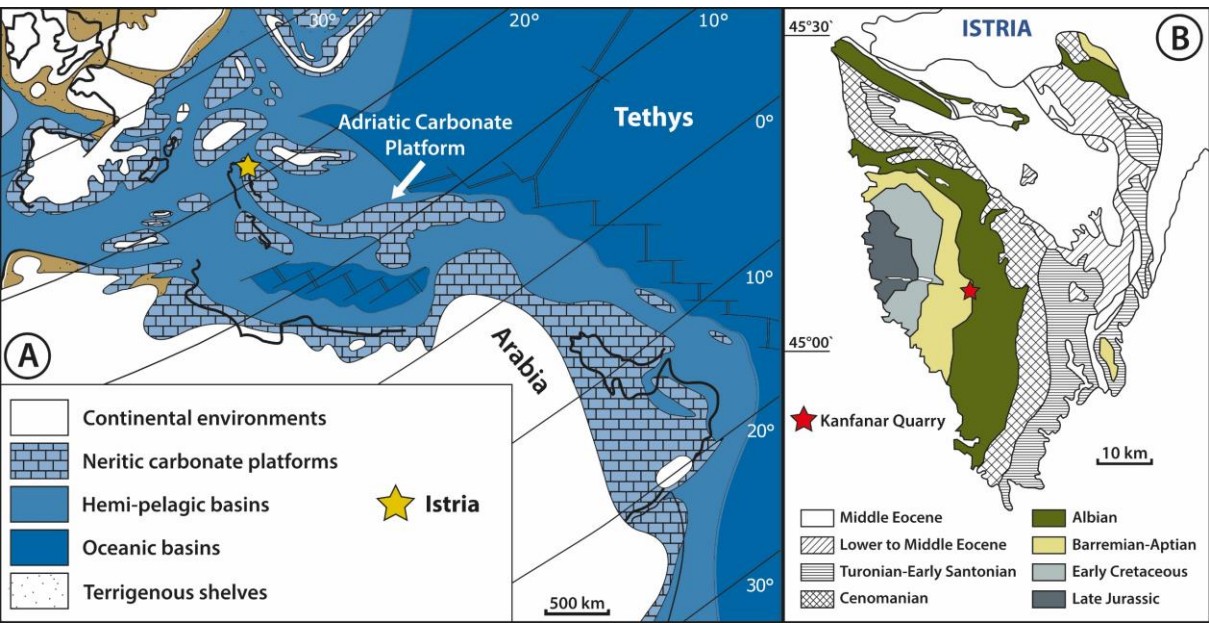

Fig. 1 (**A**) Palaeogeographic reconstruction of the early Aptian Tethyan realm (119 Ma) displaying the isolated
Adriatic Carbonate Platform surrounded by hemi-pelagic basins (approximate position of Istria is indicated with
a yellow star; modified after Masse et al., 2004 and Huck et al., 2010). (**B**) Regional geology of Istria with location
of the Kanfanar quarry (modified after Huck et al., 2010).

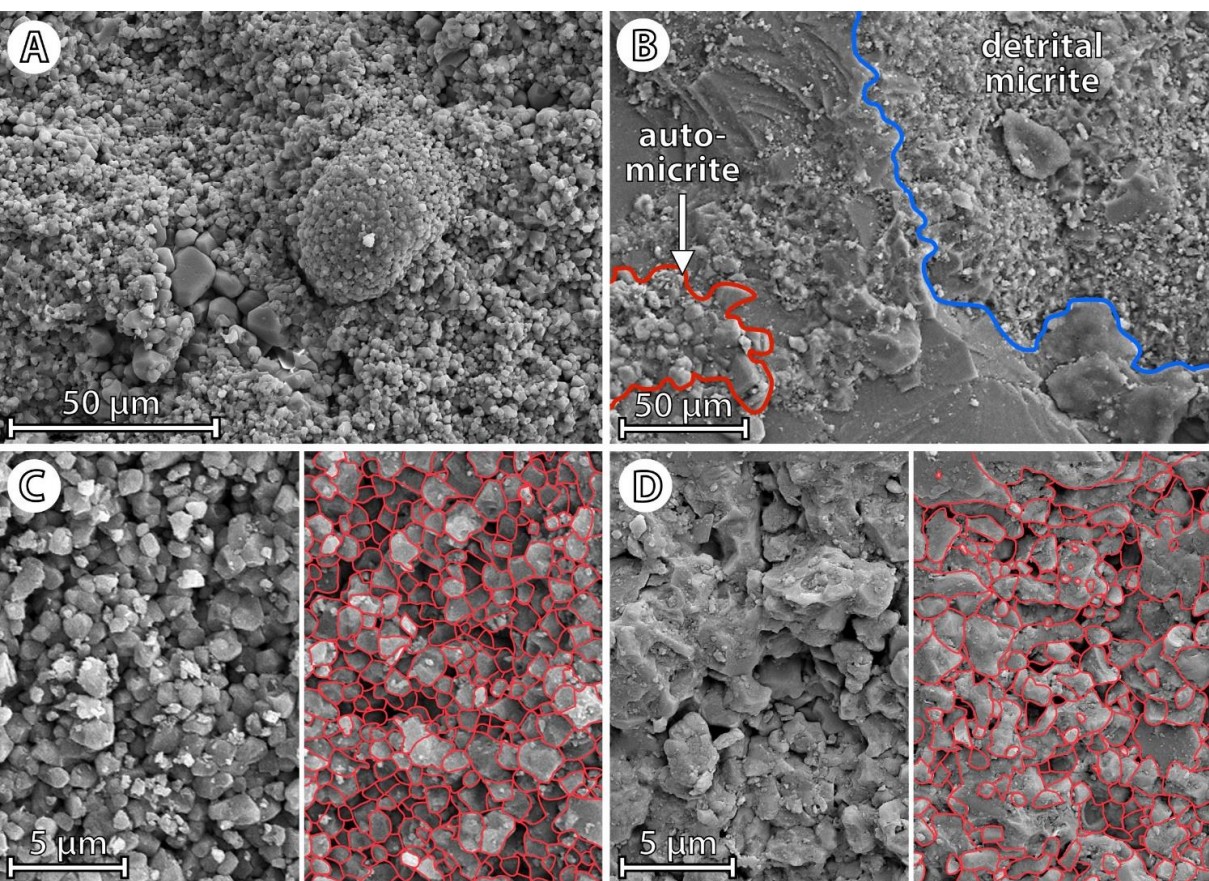

**Fig. 2.** SEM photomicrographs of automicrite *versus* detrital micrite. **(A)** and **(C)** automicrite of microencruster-dominated facies of Kanfanar Unit (22.4 m and 27.6 m; note homogenous crystal size). **(B)** Automicrite (outlined in red) and detrital micrite (outlined in blue) of the Dvigrad Unit (2.8 m). **(D)** Detrital micrite of the Dvigrad Unit (2.8 m; note variable crystal size).

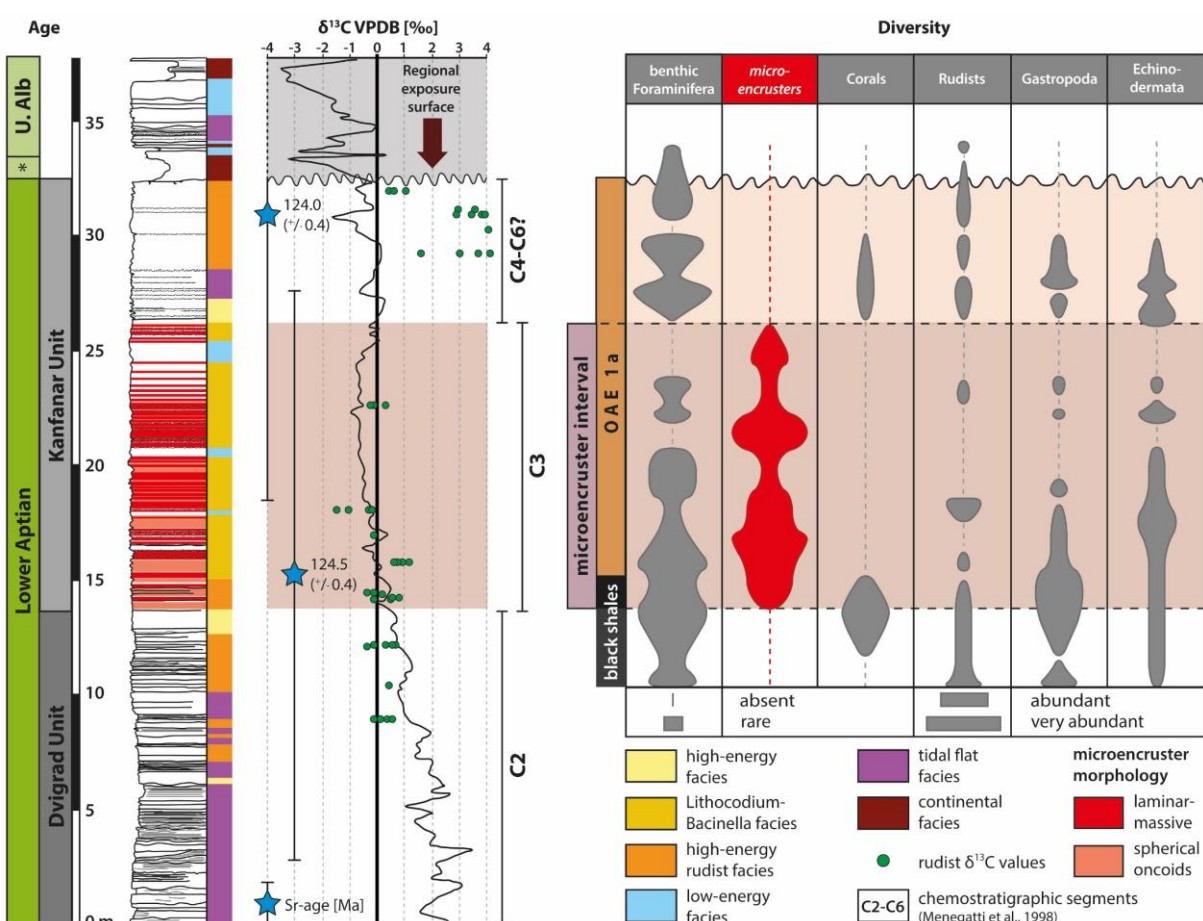

**Fig. 3.** Stratigraphic frequency distribution of marine carbonate-secreting organisms in the Dvigrad and Kanfanar units as based on thin section analysis. Stratigraphic framework and δ13C (including Sr-age) isotope stratigraphy from Huck et al. (2010). OAE1a-equivalent interval and microencruster facies are indicated. Chemostratigraphic segments (C2 through C6) are based on Menegatti et al. (1998).

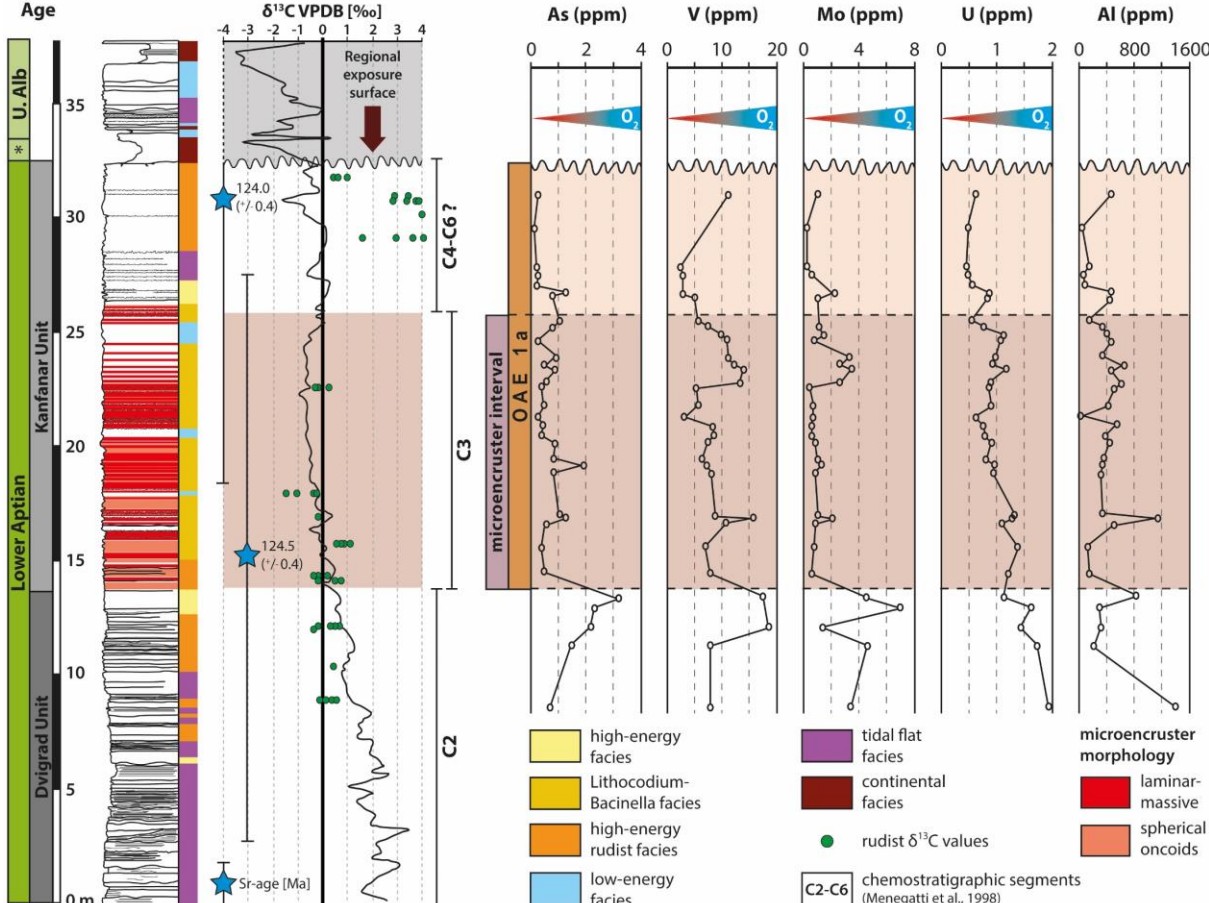

**Fig. 4.** Chemostratigraphy of redox sensitive trace elements and aluminum content in ppm across the lower Aptian Dvigrad and Kanfanar units. Note position of OAE 1a-equivalent deposits and stratigraphic interval characterized by microencruster facies (C3). Redox sensitive trace elements indicate an abrupt decrease in seawater dissolved oxygen level across the C2/C3 chemostratigraphic boundary coinciding with the onset of OAE 1a and the expansion of microencruster facies on the platform top.

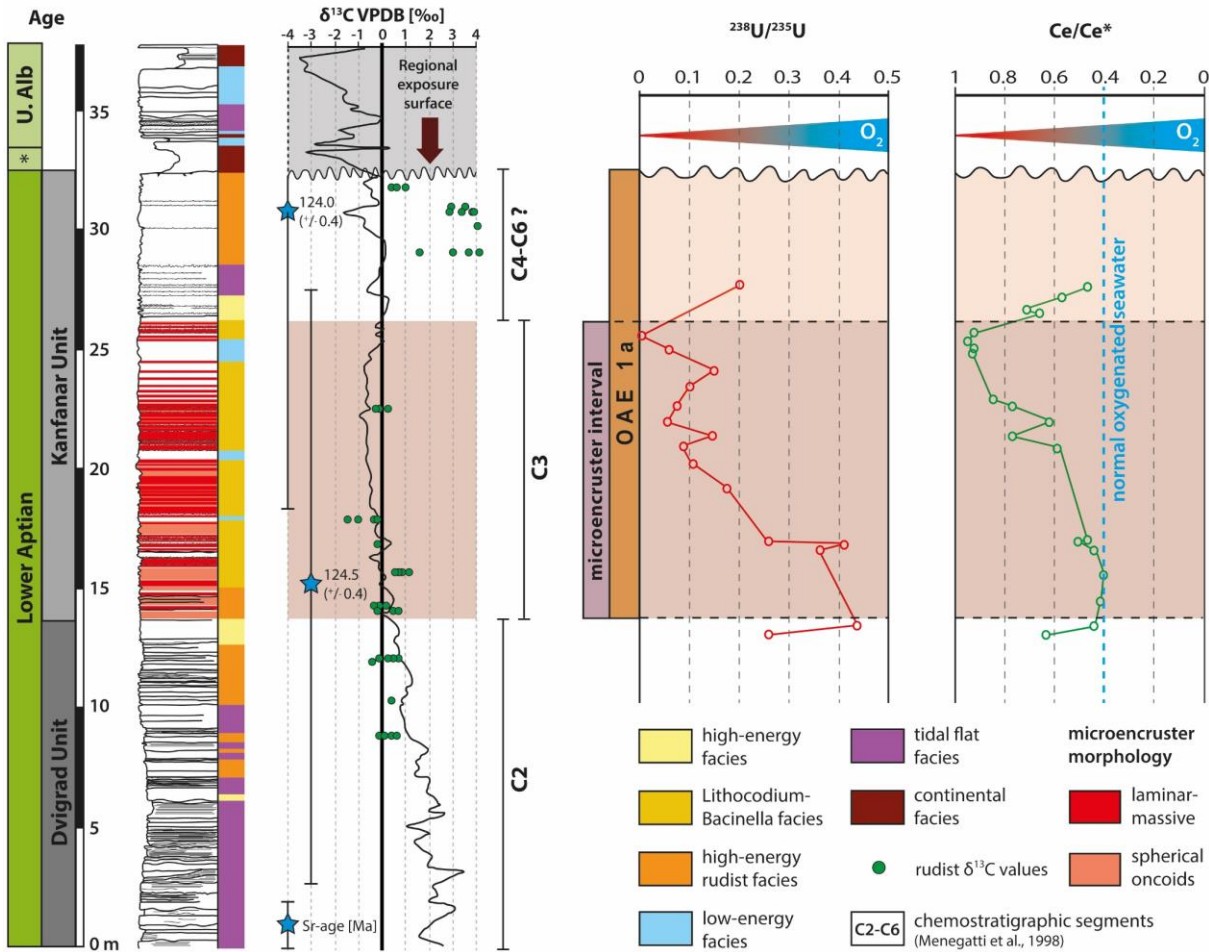

**Fig. 5.** Chemostratigraphy of $^{238}$U/$^{235}$U and Ce/Ce* ratios in the Kanfanar section. Note, decreasing U and increasing Ce/Ce* values are slightly offset during the C3 chemostratigraphic interval characterized by microencruster facies. Both redox proxies indicate a systematic decrease in dissolved seawater oxygen levels across chemostratigraphic segment 3.

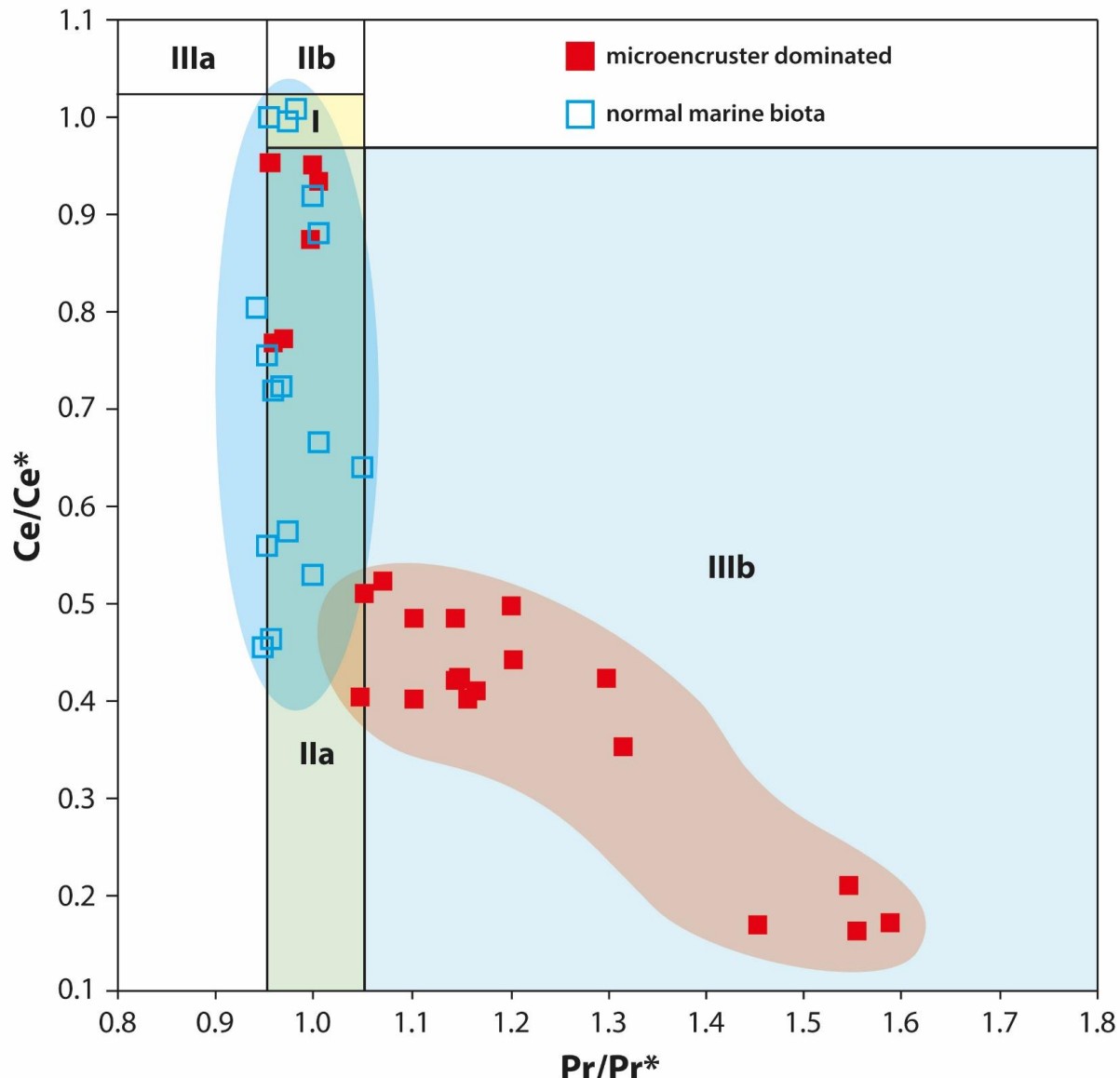

**Fig. 6.** Lanthanum anomaly diagram (Ce anomaly, y-axis *versus* Pr anomaly, x-axis). Field I: no anomaly; field IIa: positive La anomaly causing an apparent negative Ce anomaly; field IIb: negative La anomaly causing an apparent positive Ce anomaly; field IIIa: genuine positive Ce anomaly; field IIIb: genuine negative Ce anomaly. All data points plotting in field IIIb, indicating a genuine negative Ce anomaly, whereas values plotting in field IIa indicating slightly overestimated Ce anomaly values due to a positive La anomaly (after Bau and Dulski, 1996; Bodin et al., 2013).

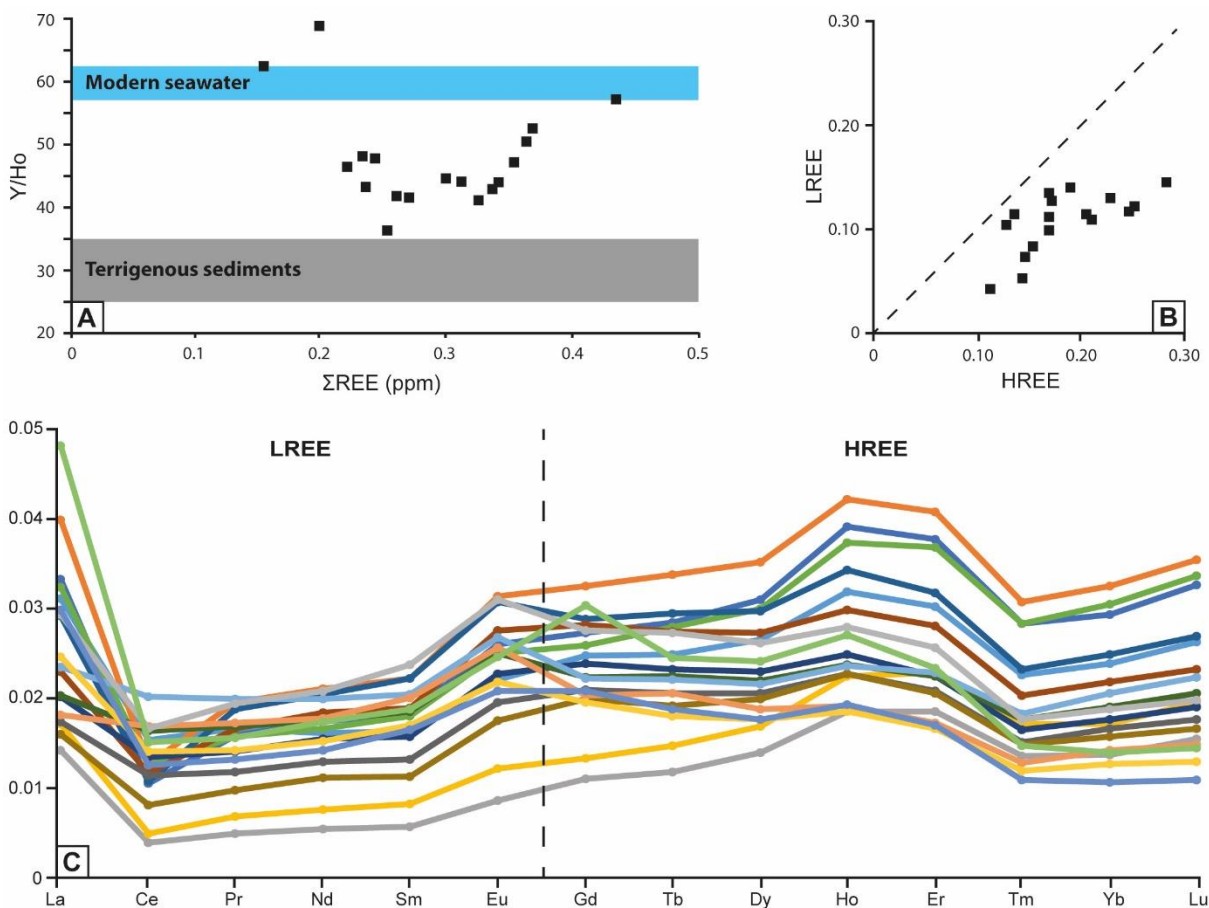

**Fig. 7.** The Y/Ho diagram **(A)** to prove for terrigenous influence, as well as Light Rare Earth Element (LREE) – Heavy Rare Earth Element (HREE) ratio **(B)**, and the REE patterns **(C)** to check for a normal marine pattern with elevated HREEs compared to LREEs.

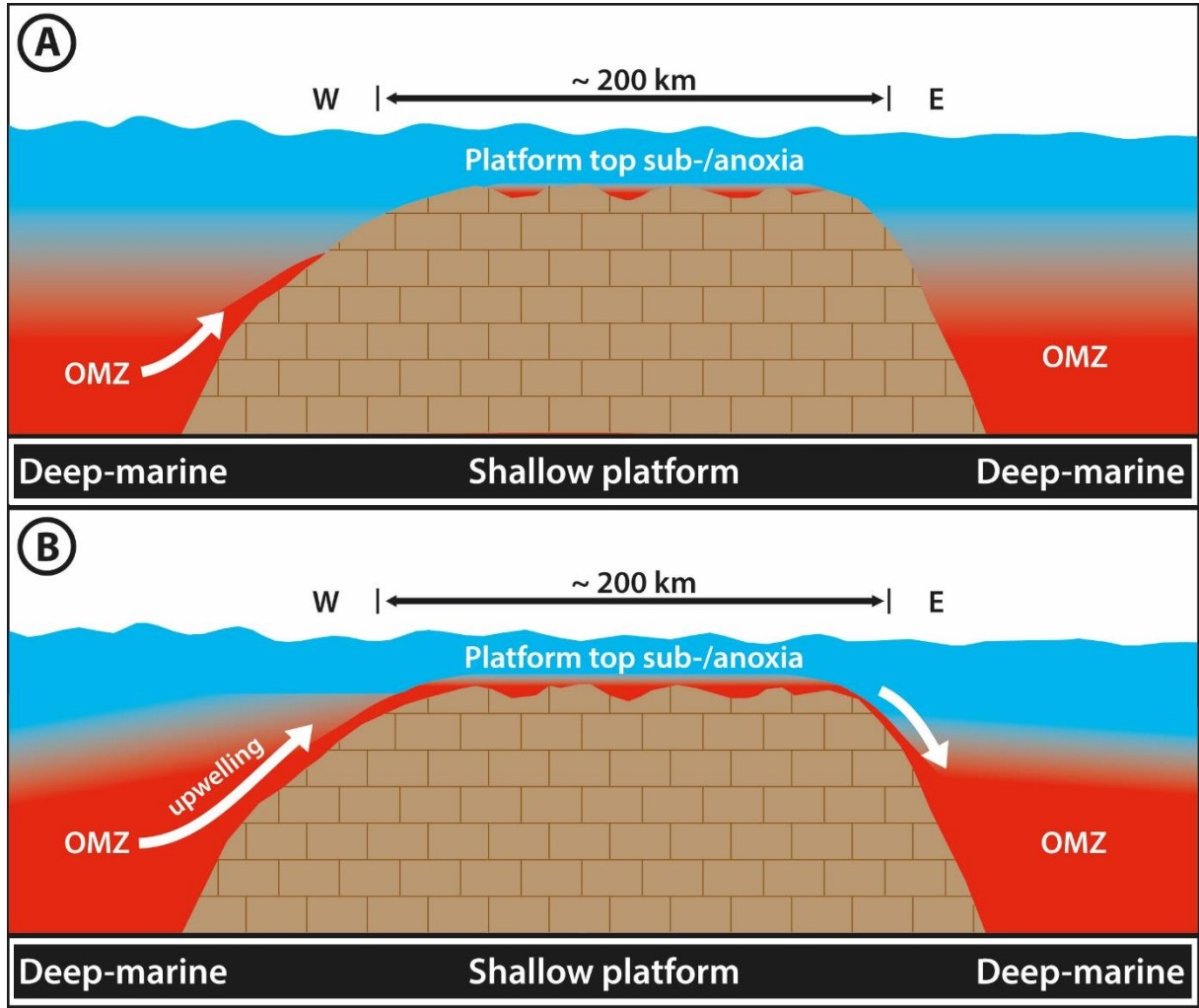

**Fig. 8.** Two models for the transient establishment of oxygen-deficient water masses on the lower Aptian Adriatic platform. **(A)** Localized seawater oxygen minimum zones form in more isolated portions of the widespread platform realm and are not connected to the oxygen depleted waters in the basin. **(B)** Transient upwelling of the oxygen minimum zone into platform water masses. Judging from recent anoxic coasts, the establishment of oxygen depleted water masses even over comparably short time intervals is sufficient to cause mortality of benthic marine ecosystems.

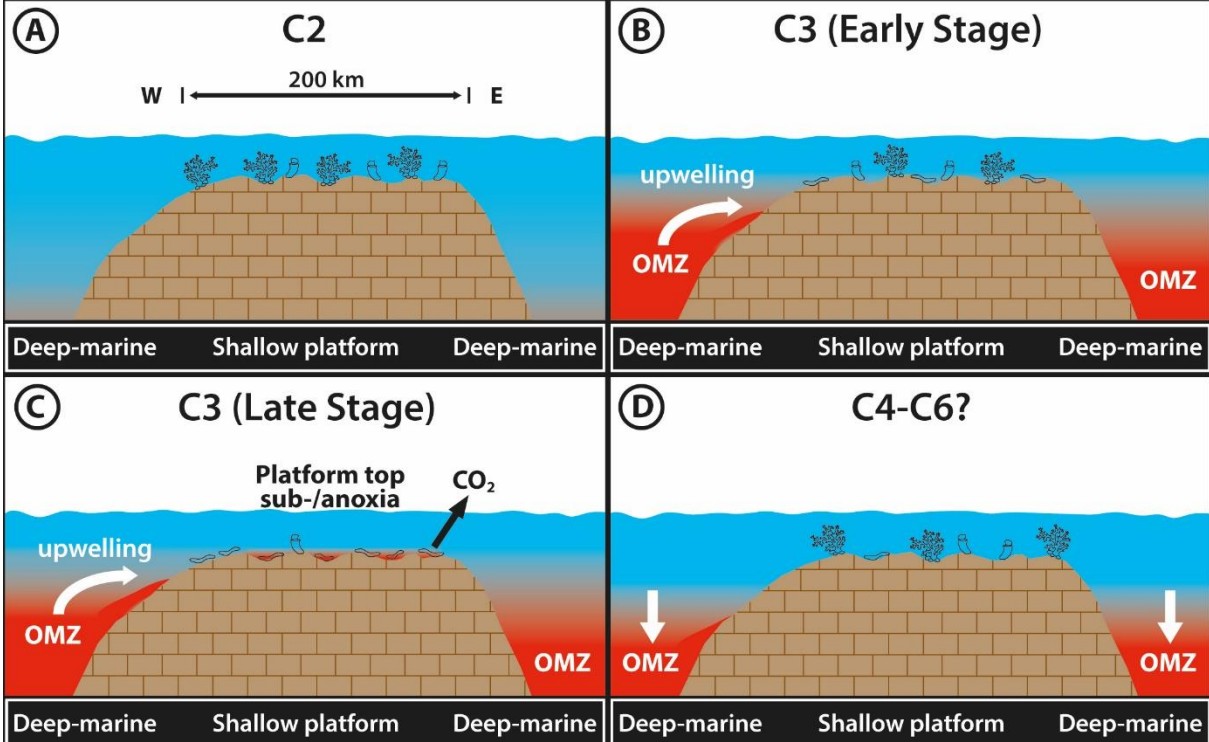

**Fig. 9.** Chronology of Adriatic platform top hypoxia subdivided in the chemostratigraphic segments C2 to C6? according to Menegatti et al. (1998). **(A)** Normal marine oxygenated conditions with only localized microencruster facies but vital coral-rudist ecosystems. **(B)** OAE1a-related oxygen depletion in basinal settings and deposition of organic-rich sediments (black shales) indicated by shift to low amounts of redox sensitive trace elements in platform carbonates due to their sequestration into deep-marine black shales. Normal oxygenated conditions prevail on the platform top. **(C)** Platform top oxygen hypoxia is established (Ce-anomaly values and U isotope ratios) and coincides with the decline of coral-rudist facies accompanied by the expansion of microencruster facies. **(D)** Return of oxygenated platform top water masses (Ce-anomaly values and U isotope ratios) and the return of normal marine benthic biota. Redox sensitive trace elements show a moderate increase, suggesting the decline of the oxygen-minimum zone in basinal settings.