# Peer review of "Central Tethyan platform-top hypoxia during Oceanic Anoxic Event 1a"

_Climate of the Past, 2019_

## Referee Comment (RC1) · Algeo (Referee) · 30 Jan 2019

This study examines geochemical proxies, mainly for watermass redox conditions, in Upper Cretaceous platform carbonates in Croatia that were deposited during OAE1a (Early Aptian). These deposits accumulated on the Adriatic Carbonate Platform, which covered an immense area (>500,000 km2) in the central Tethys during the Early Cretaceous.

The authors' main theses are that (1) redox conditions became at least somewhat reducing, possibly in multiple pulses rather than a single event, on the platform top during OAE1a, and (2) reduced oxygen levels played a role in stressing shallow-marine biotas during this event.

My opinion is that, while the presented data do not conflict with the main interpretations of this study, some of the data are open to alternative interpretations.

Detailed comments:

1) A couple of aspects of the Adriatic Carbonate Platform are not sufficiently addressed. First, what were the water depths on the platform top? Was it on the order of a few meters, like on the modern Great Bahama Bank? Presumably, the platform top was shallow if algae and corals were growing profusely on it, but this point needs to be discussed explicitly.

If the Adriatic Carbonate Platform was quite shallow, then this characteristic as well as its open margins would make the Great Bahama Bank a much more appropriate analog than the Late Pennsylvanian Midcontinent Sea of North America, which was flooded to depths of 100 m during glacio-eustatic highstands and which was nearly landlocked in its geography. The climatic boundary conditions were also quite different–in the Pennsylvanian, Gondwanan glaciation was driving large eustatic fluctuations, whereas in the Cretaceous, continental ice mass was limited, and sea-level fluctuations were small. For this reason, the Adriatic Carbonate Platform would probably have built up to close to sea level. (I thank the authors for making use of my work on the LPMS, but it is probably not the best choice of analog.)

Second, the study site is shown at a paleolatitude of 22 degrees N. This is squarely in the dry subtropics, so there was probably strong evaporation over the top of a large platform like this. If platform-top water depths were quite shallow, then evaporation may have been strong enough to make these waters significantly more saline than normal seawater. This might have set up off-platform flow of a hypersaline watermass, which would have had consequences relevant to the interpretations of the present study (e.g., a mechanism to drive lateral advection or upwelling of open-ocean waters onto the platform top) as well as to topics that are not considered here (e.g., formation of warm saline deep waters in the Tethys).

2) The interpretations of redox proxies in this study are not incorrect, but they are incomplete. Additional considerations that need to be discussed in the manuscript include the following:

First, several redox-sensitive trace elements (As, V, and Mo) were used to evaluate redox variations. The elemental data were generated with LA-ICP-MS, and presumably the authors targeted carbonate samples and steered away from shaly samples. If the samples are 100

If the study samples are not pure carbonate, then raw trace element concentrations are not particularly meaningful unless an Al concentration curve is plotted next to them, or the trace element concentration profiles are Al-normalized. At various points in the manuscript, increases in trace element concentrations are attributed to more reducing conditions (e.g., Page 7, Line 22), but what if they just reflect an increase in the clay content of samples? The data needed to test this possibility have not been supplied.

Because redox-sensitive trace elements are local redox proxies, they will reflect platform-top conditions (i.e., at the site of deposition), not conditions in adjacent basinal areas, so the statement that "Redox sensitive elements still display low concentrations, pointing to ongoing oxygen depletion in the basin" (Page 11, line 37) seems suspect.

Second, cerium anomalies (Ce/Ce*) have been widely used as a paleoredox proxy, but too many studies have failed to demonstrate that the measured REEs represent a hydrogenous source (i.e., seawater-derived) and not just a detrital source. If the REEs in a sample are mainly detrital in origin, then they cannot comment on depositional redox conditions. The sources of REEs can be readily tested using two approaches: (1) Y/Ho ratios, and (2) LREE/HREE ratios. Y/Ho ratios are 25-30 for REEs in clay minerals, but typically >60 for REEs in seawater. The relative amounts of REEs drawn from lithogenous versus hydrogenous sources can be estimated from how closely Y/Ho ratios in the study samples approach these endmember values. Furthermore, REE distributions are roughly flat in clay-rich samples (i.e., LREE/HREE close to 1.0; although

there may be strong MREE enrichment if phosphate is present), but they will show LREE depletion (usually with a flattish MREE-HREE distribution) in samples containing hydrogenous REEs. The theory for these applications was discussed in Chen et al., 2015 [Chen, J., Algeo, T.J., Zhao, L., Chen, Z.Q., Cao, L., Zhang, L. and Li, Y., 2015. Diagenetic uptake of rare earth elements by bioapatite, with an example from Lower Triassic conodonts of South China. Earth-Science Reviews, v. 149, pp. 181-202]. Before Ce/Ce* can be meaningfully used as a paleoredox proxy, it is necessary to first demonstrate that the REEs in a sample are mainly of hydrogenous origin based on these two tests. I suspect that the REEs in the present study samples are of hydrogenous origin, but this needs to be demonstrated rather than surmised.

Third, the various redox proxies utilized in this study seem to be treated as equivalent to each other, but this is not at all the case–they have different temporal and spatial scales of significance owing to differences in seawater residence times. Carbonate U-isotopes represent a global-ocean redox proxy owing to the long residence time of U in seawater ( 450 kyr). At the other end of the spectrum, redox-sensitive trace metals are taken up locally across the sediment-water interface, and thus respond directly to local redox conditions (although also indirectly to basinal or global redox conditions through redox-related changes in their seawater inventory). Ce/Ce* falls in between owing to a residence time in seawater of one to a few hundred years–which is long enough to record basinal effects but generally not long enough to record global effects owing to the 1000-2000 yr mixing time of the oceans. Thus, the redox proxies utilized in this study are providing different types/scales of redox information. The carbonate U-isotope and Ce anomaly proxies show similar secular patterns, suggesting that both are recording basin-scale shifts in redox conditions–they both appear to record a single large shift toward more reducing conditions during OAE1a. The trace-element proxies do not show significant covariation with the U and Ce proxies, and their largest peaks are below the OAE1a interval. However, their concentrations are so low (mostly a few ppm) that it is unclear whether they are recording redox variations at all (as opposed to clay content; see above). This is a critical issue for the interpretations of the

present study because the U and Ce proxies are likely to be recording a shift toward more reducing conditions in the larger Tethys Ocean basin, and not on the platform top per se–only the redox-sensitive trace elements provide information specifically about platform-top redox conditions, and if their concentrations are controlled by clay content, then there is no evidence for changes in redox conditions on the platform top during OAE1a. Indeed, this aspect of the interpretations of this study must be doubted, as it would be quite difficult to make a watermass that is only a few meters deep oxygen-depleted–such waters simply exchange gases too freely with the atmosphere.

3) Some terminology used in this study is non-standard, specifically the term "automicrite" (page 6). This term is not defined or described explicitly anywhere in the manuscript. Reading 'between the lines', I interpret it to mean "authigenic micrite", i.e., micrite precipitated in situ. If this is this case, however, then it is effectively a micritic cement, formed either on the seafloor or within sediment pore spaces–and it would be clearer to call it "micritic cement".

Two issues that must be addressed are: (1) what is the exact origin of "automicrite"?, and (2) what are the petrographic or field criteria used to distinguish this type of micrite from "detrital micrite"? In this regard, the authors appear to assume that the "automicrite" in their samples is a primary component, and not recrystallized material, but the images provided do not permit a definitive interpretation on this point. Basically all fine-grained carbonate recrystallizes in the burial environment, sometimes multiple times, and the recrystallization effects can be highly localized (e.g., see Algeo et al., 1992: [Algeo, T.J., Wilkinson, B.H. and Lohmann, K.C., 1992. Meteoric-burial diagenesis of Middle Pennsylvanian limestones in the Orogrande Basin, New Mexico; water/rock interactions and basin geothermics. Journal of Sedimentary Research, v. 62(4), pp. 652-670.] In the present study, Turpin et al. (2012) is cited as the source of "further criteria" for automicrite, but any relevant criteria need to be elaborated upon here.

4) The second goal of the study is to show the influence of reduced oxygen levels in stressing shallow-marine biotas during the OAE1a event. However, establishing

the causation of biotic events is always quite difficult. Even assuming that there was unambiguous evidence of a shift toward reducing conditions on the platform top (which is not entirely clear; see above), how is it possible to show that changes in redox conditions and not in some covariant factor (such as water temperature) was the main biotic stressor?

Minor matters:

1) The manuscript contains many instances of past events being described in the present tense. Past events should be described using the past tense. 2) Page 2, line 31: what is an "out-of-balance reefal ecosystem"? The intended meaning is unclear here. 3) Page 6, line 18: what is the evidence for an "increased sedimentation rate"? 4) Some grammatical problems, e.g., page 11, line 32: "portions ... coincide" –need to have subject-verb agreement.

Thomas Algeo University of Cincinnati State Key Labs of GPMR and BGEG, China University of Geosciences 29 January 2019

---

## Short Comment (SC1) · 30 Jan 2019

The paper by Hueter et al. is an interesting contribution to better understanding the environmental impact of the oceanic anoxic event (OAE) 1a on shallow-marine carbonate platforms. However, there are a few problems with the data presentation and interpretation that merit comment as well as the paper overlooks an earlier paper with a more detailed modified whole rock $\delta$13C curve and facies work that documents hypoxia in southern Croatia (Husinec et al., 2012, 2018) and suggests that it occurred slightly later than hypothesized by the present authors.

The overall shape of the bulk carbonate matrix-based $\delta$13C curve (Fig. 3) does not allow definite designation of the Menegatti et al. (1998) C-isotope segments. The

$\delta$13C curve from Huck et al. (2010; their Fig. 10) has been modified by significantly shifting the segment C3 boundaries: the base of C3 is shifted $\sim$8 m higher in the section (i.e., from the upper Dvigrad to the base of Kanfanar Unit), and its top is now picked on top of the last microencruster occurrence, $\sim$7 m higher in the section (i.e., shifted from lower to upper Kanfanar Unit). Thus, they have significantly shifted their C3 pick up-section. The authors should acknowledge that and explain what the new picks are based on. The +/- 0.4 my error bars on the Sr-isotope ages are much larger than the interval the authors are trying to date – we suggest you put the error bars on the figure.

A comparison between the $\delta$13C curve with all Menegatti et al. (1988) segments has previously been published for the southern Adriatic Platform by Husinec et al., (2012, 2018) and should be discussed. The overall shape of that curve, unlike the Huck et al. (2010) incomplete curve (segments C2 to C6?) used in the current study, suggests correlation with classic pelagic sections (e.g., Vocontian Basin - Föllmi et al., 2006) that places the hypoxic interval (shown in light grey as deeper lagoon facies on Fig 8) slightly younger than in the present paper. In fact, our S Adriatic $\delta$13C curve (Husinec et al., 2012) closely resembles the Oman and SE France curves which clearly delineate the C3 segment. In Oman, the Lithocodium-Bacinella interval spans from uppermost C3 to C6 segments, similar to our dysaerobic laminated interval that is barren of any fauna and spans C4 to perhaps C7. However, if the relative ages between the Istria and southern Adriatic Platform do indeed differ, then it might suggest diachroneity in timing of hypoxia, perhaps due to differential warping of the platform.

The final and probably the most important comment is related to an episode of platform-top hypoxia during the OAE1a (their inferred C3 segment) that the authors nicely documented using the redox-sensitive trace elements and the cerium anomaly (Figs. 4, 5). The authors then suggest return to oxygenated platform-top waters during segments C4-C7 (Fig. 8D). There are several problems with this interpretation: (1) The studied section above C3 is designated as "C4-C6?", suggesting that the designation

of the C6 top is tentative. It is then not clear what makes the authors suggest that there was no hypoxia on the platform during stage C7 (the one that is not present at all in their section)? (2) Ce/Ce* indeed shows return to normal oxygenated water (Fig. 5); however, this trend is present only within the approximately lower $\frac{1}{4}$ of their segment "C4-C6?". No Ce/Ce* samples in the remainder of the segment to suggest changes in the platform-top water oxygenation levels; (3) the authors state that the "segment C4 is characterized by patterns in redox sensitive proxies typical of normal marine dissolved oxygen levels". First of all, and as previously mentioned, Figures 4-5 show the segment "C4-C6?", i.e., based on their $\delta$13C curve, the authors can't pick the upper boundary of the segment C4. Moreover, the lower part of that same segment "C4-C6?", which could as well represent C4 only, shows the similar or even lower redox-sensitive trace-element values (As, V, Mo), thus suggesting possible continuation of hypoxia, not return to normal oxygen levels. Interestingly, in southern part of the Adriatic Platform, Husinec et al., (2012, 2018) have documented a 10-m-thick interval of platy, planar-laminated, fine pelletal lime mudstone that formed under dysoxic conditions, as evidenced by its texture, lack of any fossils and/or bioturbation, very dark gray to black color, and distinct petroliferous odor. This low-oxygen (hypoxia?) OAE1a platform equivalent appears to span from the upper part of C4 to the top of C7 (Husinec et al., 2012; Fig. 5), and may have filled local structural downwarps on the platform. The bottom line is that the Adriatic platform-top hypoxic events were not limited to the C3 segment of the OAE1a, as suggested by the authors based on the data from the NW part of the platform. The apparent younger development of hypoxia on the southern Adriatic platform during C4-C7 suggests that Fig. 8D needs to be modified or the C3 age of the hypoxia re-evaluated.

Sincerely,

Antun Husinec and J. Fred Read

References:

Föllmi, K.B., Godet, A., Bodin, S., Linder, P., 2006. Interactions between environmental change and shallow-water carbonate buildup along the northern Tethyan margin and their impact on the Early Cretaceous carbon-isotope record. Paleoceanography 21, PA4211.

Huck, S., Rameil, N., Korbar, T., Heimhofer, U., Wieczorek, T.D. and Immenhauser, A., 2010. Latitudinally different response of Tethyan shoal-water carbonate systems to the Early Aptian Oceanic 5 Anoxic Event (OAE 1a). Sedimentology 57, 1585–1614.

Husinec, A., Harman, C.A., Regan, S.P., Mosher, D.A., Sweeney, R.J., Read, J.F., 2012. Sequence development influenced by intermittent cooling events in the Cretaceous Aptian greenhouse, Adriatic platform, Croatia. American Association of Petroleum Geologists Bulletin 96, 2215–2244.

Husinec, A., Read, J.F., 2018. Cyclostratigraphic and $\delta$13C record of the Lower Cretaceous Adriatic Platform, Croatia: Assessment of Milankovitch-forcing. Sedimentary Geology 373, 11–31.

Menegatti, A.P., Weissert, H., Brown, R.S., Tyson, R.V., Farrimond, P., Strasser, A., Caron,M., 1998. High-resolution $\delta$13C stratigraphy through the early Aptian "Livello Selli" of the Alpine Tethys. Paleoceanography 13, 530–545.

---

## Author Comment (AC2) · 11 Feb 2019

Dear Dr. Algeo,

thank you very much for carefully reading our manuscript and your detailed and insightful comment. This discussion will significantly help to improve the quality of this study. Please find attached our step by step reply to your comments.

Sincerely yours,

A. Hueter on behalf of all co-authors.

Please also note the supplement to this comment:
https://www.clim-past-discuss.net/cp-2019-3/cp-2019-3-AC2-supplement.pdf

---

## Referee Comment (RC2) · Helmut Weissert (Referee) · 26 Mar 2019

The authors of this paper aim to test the hypothesis that low oxygen concentrations in coastal waters at times of widespread Cretaceous ocean anoxia (OAE's) had an impact on carbonate platform growth during times of C-cycle perturbations. The authors clearly formulate their hypothesis based on actualistic examples from a variety of coastal regions. Enigmatic microencruster blooms in Tethyan shallow water carbonate successions of Cretaceous age have repeatedly been described by numerous authors as correctly mentioned by the authors of this study. Therefore, a new and detailed investigation of causes resulting in observed changes in carbonate platform ecology seems timely and of relevance. The geochemical methods the research team is applying for reconstruction of oxygen levels are well explained and the authors seem

to be aware of problems related to "contamination" of their carbonate samples (clay input, for example) altering carbonate geochemistry of their samples. In the following paragraphs, I like to add some comments which may help to further improve this manuscript.

1) The authors investigated a shallow water carbonate succession formed on the vast Cretaceous Adria carbonate platform. This section today is outcropping in Croatia. Stefan Huck and colleagues had published earlier studies, including stratigraphic investigations, on this section. Therefore, the authors of this paper only summarize stratigraphic information relevant for this study. If I look at the C-isotope stratigraphy presented in this paper, based on earlier studies, I still feel somewhat uneasy with the stratigraphic interpretation made by the authors. I like to refer to another, more recent study from Adria platform made by Sabrina Amodio and myself (2017). In this study (and in earlier ones by the Napoli groups) from the S. Apennines we succeeded in identifying the base of the Aptian positive excursion and the prominent short negative spike preceding this excursion. This pattern is not seen in the section from Croatia, either due to diagenesis or due to gaps in the record (?). In addition, we found a microencruster level in the Lower Aptian carbonates preceding OAE1a which possibly may be correlated with the level described in this study (?). I recommend that the authors have a look at this study and that they evaluate the possibility that the studied microbial level is of pre-OAE1a age. Is it possible that the section in Croatia corresponds to the Barremian-Aptian interval with microencrusters documented by Amodio and Weissert (e.g. Monte Faito ?). Even if the authors would come to the conclusion that the section may represent a pre-Aptian interval, the questions and related problems formulated by the authors remain of equal relevance. Numerous black shale deposits documented from deep Tethyan sections were formed also during pre-OAE1a times. It may, therefore, be hypothezised, that one of these pre-OAE1a anoxic events coincided with the prominent microencruster level described in this study.

2) Proxies - anoxia The authors present their data in nice graphic displays. Of course,

the reader immediately recognizes that correlation between different proxies is not at all straightforward. Ce and U-isotopes seem to co-vary. However, the low oxygen levels inferred from these data do not perfectly well match with the microencruster facies. The authors refer in their text that low oxygen levels indicated by Ce-anomalies and U-isotopes coincide with peak encruster occurrence at the top of their C-3 interval (P 8 line 510). Please mark this peak episode in encruster occurrence in your graph. Differences in anoxia indicators deserve a more detailed discussion (local vs regional vs global signal).

3) Some minor comments p.2 Line 23 add Wissler et al., 2003 (first detailed documentation of C-isotopes and carbonate crises) p.7 line 15 (2at 6m) what do you mean? p.8 line 38 > there is some evidence from Nd isotope data, that a possible source of deep water in the Tethys (Late Cretaceous) was on the extended carbonate platforms > downwelling? or upwelling ? You may cite one of these studies, even if they give in formation on the Late Cretaceous (e.g. Martin et al., 2012). p. 11, line 23 There were abundant black shales also in the Barremian (Weissert et al., 1979, Bersezio 1993, Giorgioni et al., 2015), just to give some examples. These regional anoxic episodes could, in agreement with your hypotheisis, also have had an impact on shallow water conditions. p11, line 32 you may cite Méhay et al., 2009, this biomarker study provides detailed information on changes in pCO2atm at the base of OAE1a.

I consider the paper as a very valuable contribution to the discussion on conditions resulting in the repeated occurrence of microencruster facies in Barremian-Aptian shallow water carbonate successions. Integration of data from other available sections from Adria (Croatia, S. Apennines) in the discussion is recommended and this integration of other data may result in a critical discussion of stratigraphy and the difficulties in using chemostratigaphy in the sediments studied.

―――――――――――――――――――――――

---

## Author Comment (AC3) · 28 Mar 2019

Dear Dr. Weissert,

Thank you very much for carefully reading our manuscript and your detailed and insightful comments.

RC#1: Stratigraphic interpretation of the Kanfanar section

AR#1: As you already noticed, the biostratigraphy of the Kanfanar section is based on the occurrence of benthonic foraminifera (Palorbitolina lenticularis) as established in a previous study by Huck et al. (2010). We are aware of the difficulties regarding the stratigraphy of shallow-water carbonates and open for any further interpretations. The carbon-isotope pattern established for the Monte Faito section by Amodio and Weis-
sert (2017) reveals many similarities with the Kanfanar carbon-isotope record. Having a closer look at the $\delta$13C pattern of the Monte Faito location, the positive peak of +4 ‰ (section meter 86) fits nicely with the +4 ‰ positive excursion (around section meter 30) in the Kanfanar section (rudist data). If this assumption is correct, the mass occurrence of microencrusting organisms (Lithocodium aggregatum and Bacinella irregularis) would not represent the Selli Level Equivalent, the latter being masked by a regional exposure surface higher up in the Kanfanar section (at section meter 32.5). This would lead to the interpretation of a pre-OAE 1a age for the studied microencruster interval in the Kanfanar section.

On the other hand, Huck et al. (2010) compared the $\delta$13C pattern of the Kanfanar section with the patterns of a composite section in Oman and the La Bedoule Section in France, presenting many similarities as well. These similar patterns share the strong positive peak in $\delta$13C of +4 ‰ (rudist shells in the Kanfanar section) and would rather support the current interpretation of the stratigraphic framework, additionally encouraged by Sr ages (Oman and Croatia).

Nevertheless, we are thankful for this helpful interpretation to this apparently very complex topic and will not exclude this possibility, due to the already known occurrences of L. aggregatum and B. irregularis before and after OAE 1a (e.g. Portugal and Oman). As you already stated, the connection of hypoxia or even anoxia and the massive occurrence of microencrusting organisms would not be affected by this new interpretation. Anyway, the biostratigraphy and the possible connection between OAE 1a and microencruster occurrences is very important for the understanding of these complex cause and effect patterns. Therefore, we will discuss this possibility in detail in a revised version of this manuscript.

RC#2: Maximum of oxygen depletion (as seen in uranium isotope ratios and Ce-anomaly values) coincides with the maximum abundance of microencrusting organisms at the top of chemostratigraphic segment C3.

AC#2: In your comment, you are referring to text that states that the maximum oxygen depletion (as seen in the geochemical proxies) coincides with the maximum abundance of microencrusting organisms at the top of chemostratigraphic segment C3. We agree that this statement is somehow misleading, as figure 3 shows the maximum abundance at section meter 17 and 22. Indeed, the abundance is already decreasing at the top of C3. The text should probably better read: At the top of the C3 segment, uranium isotope ratios and cerium anomalies indicate a maximum oxygen depletion, but the abundance of microencrusters is already decreasing. There are two possible explanations for this: (1) Regarding the complex morphology of L. aggregatum and B. irregularis (e.g. Rameil et al., 2010), the two-dimensional perspective in thin sections may induce some degree of data bias, i.e. the thin section area covered by microencrusters strongly depends on the plane of observation. Moreover, the abundance and the grown morphology may change due to changing environmental parameters. We are aware of this problem and took great care to compare observations from the field and thin sections observation to reduce any potential error to a minimum. Despite all our care, we cannot exclude, however, that some of the second order patterns are not, to some degree, influenced by these issues. In short, this is a complex, non-perfect, geological system and any sampling strategy may induce this type of problems. We are confident that the first order patterns are valid and can be replicated though. (2) The more likely interpretation is: The oxygen depletion is reaching its maximum at the top of C3 and becomes critical for even the hardiest organisms, exceeding their tolerance limits and hence they decline. We will change the relevant statements in the manuscript and discuss the possible explanations for the decrease in microencruster abundance despite persistent oxygen deficiency.

As a second important point of your second comment refers to the discussion of our geochemical redox proxies. We agree that uranium isotope ratios will preserve a global signal due to the long residence time of uranium in the global ocean (450 kyr). In contrast, cerium, with a residence time of a few hundred years, as well as the redox sensitive trace elements, will display a local or regional signal (Sholkovitz and Schneider, 1991; Shields and Stille, 2001; Bodin et al., 2007), supported by the long mixing time of the oceans (∼1000-2000 years). For this reason, we will add some text in the discussion that deals with the temporal level on which the redox proxies operate.

Minor comments: (1) P.2, line 23: We will add Wissler et al. (2003) as a further reference. (2) P.7, line 15: It should be "(at 26 m)" instead of "(2at 6 m)". Will be changed in a revised version. (3) P.8, line 38: We will add some new references and discuss the question of upwelling or downwelling. (4) P.11, line 23: We will discuss the possible impact of Barremian black shale deposition on the studied interval. (5) P.11, line 32: We will cite Méhay et al. (2009) to provide detailed information about the changes in pCO2 at the base of OAE 1a.

Thank you for you very constructive comments.

Sincerely yours, A. Hueter on behalf of the co-authors.

References:

Amodio, S. and Weissert, H. (2017): Palaeoenvironmental and palaeoecology before and at the onset of Oceanic Anoxic Event (OAE) 1a: Reconstructions from Central Tethyan archives. Palaeogeography, Palaeoclimatology, Palaeoecology, 479, 71-89. http://dx.doi.org/10.1016/j.palaeo.2017.04.018.

Bodin, S., Godet, A., Matera, V., Steinmann, P., Vermeulen, J., Gardin, S., Adatte, T., Coccioni, R. and Föllmi, K.B. (2007): Enrichment of redox-sensitive trace metals (U, V, Mo, As) associated with the late Hauterivian Faraoni oceanic anoxic event. Int. J. Earth. Sci., 96, 327-341. https://doi.org/10.1007/s00531-006-0091-9.

Huck, S., Rameil, N., Korbar, T., Heimhofer, U., Wieczorek, T.D. and Immenhauser, A. (2010): Latitudinally different response of Tethyan shoal-water carbonate systems to the Early Aptian Oceanic Anoxic Event (OAE 1a). Sedimentology, 57, 1585-1614. https://doi.org/10.1111/j.1365-3091.2010.01157.x.

Méhay, S., Keller, C.E., Bernasconi, S.M., Weissert, H., Erba, E., Bottini,

C. and Hochuli, P.A. (2009): A volcanic CO2 pulse triggered the Cretaceous Oceanic Anoxic Event 1a and a biocalcification crisis. Geology, 37(9), 819-822. https://doi.org/10.1130/G30100A.1.

Rameil, N., Immenhauser, A., Warrlich, G., Hillgärtner, H. and Droste, H.J. (2010): Morphological patterns of Aptian Lithocodium-Bacinella geobodies: relation to environment and scale. Sedimentology, 57, 883-911. https://doi.org/10.111/j.1365-3091.2009.01124.x.

Shields, G. and Stille, P. (2001): Diagenetic constraints on the use of cerium anomalies as palaeoseawater redox proxies: an isotopic and REE study of Cambrian phosphorites. Chem. Geol., 175, 29-48. https://doi.org/10.1016/S0009-2541(00)00362-4.

Sholkovitz, E.R. and Schneider, D.L. (1991): Cerium redox cycles and rare earth elements in the Sargasso Sea. Geochim. Cosmochim. Acta, 55, 2737-2743, https://doi.org/10.1016/0016-7037(91)90440-G.

Wissler, L., Funk, H. and Weissert, H. (2003): Response of Early Cretaceous carbonate platforms to changes in atmospheric carbon dioxide levels. Palaeogeography, Palaeoclimatology, Palaeoecology, 200, 187-205. https://doi.org/10.1016/S0031-0182(03)00450-4.

―――――――――――――――――――――――

---

## Author Response (AR3)

[revised manuscript text omitted]

Dear colleagues,

Thank you very much for your detailed and insightful comment. We believe, this is really the essence of a discussion version of a paper and we welcome your comments. Please find our reply below:

Comparing the $\delta^{13}C$ curve of Huck et al. (2010) and the one of Husinec et al. (2012; 2018), and taking the $\delta^{13}C$ of rudists from Huck et al. (2010) into account suggests many similarities (which is good). Due to the geographical distance between the Kanfanar and Mljet sections, there might be potentially different burial and generally, diagenetic, histories that must be taken into consideration. In our view, the bulk $\delta^{13}C$ signatures in the Mljet section have seen less diagenetic alteration. In your comment, you refer to the fact that we have shifted the C3 and C4 segments relative to the Huck et al. (2010) paper. Please allow us to explain why we think that this is justified: Answered on page 12, lines 13-21 in the revised manuscript.

In our view, the first occurrence of microencrusting organisms in the platform domain is coeval with the onset of hypoxic conditions in basinal settings expressed as organic-rich deposits comparable to black shales. The C3 segment is characterized by a pronounced negative $\delta^{13}C$ excursion and marks the onset of OAE 1a equivalent in Kanfanar. When comparing this pattern with the Cismon section (Menegatti et al., 1998), there defined by the first occurrence of black shales, evidence for the assignment of the C3 segment in Kanfanar to the base of the Kanfanar Unit is found. The top of C3 as well as the base of C4 in Kanfanar are picked on top of the last microencruster occurrence based on the lowest seawater oxygen content, as suggested in Ce/Ce* and by means of redox sensitive trace element concentration, as well as the $\delta^{13}C$ signatures of rudist low-Mg calcite as shown in Huck et al. (2010). Answered on page 7, line 36 and on page 8, lines 1-10 in the revised manuscript.

Are these statements in disagreement with the comments made by Husinec and Read? We believe not! We feel that assuming a high degree of temporal correlation between two sections situated perhaps 1000 km apart, is not likely. Actually, our view of a "global" event is that of a global clustering of regional (and even local) events that are all more or less "coeval". Answered on page 12, lines 13-21 in the revised manuscript.

Finally, let us comment on the issue of the C4-C7 segments in figure 8. We fully agree that the assignment of the C4-C6 intervals (for example in Fig. 4) are less well constrained. Hence, no disagreement here. Please note the question mark (for example figure 4 etc.). You suggest that you record ongoing hypoxia into segment C7. We are happy with this and again refer to the fact that we should not assume a fully coeval expression of seawater oxygen level across a large, topographically complex platform. Please also note that we documented significant regional differences in the timing of the first and last occurrence of microencrusters providing solid evidence of the complexity of a system that sees a combination of global, provincial and regional environmental patterns (perhaps similar to sea-level reconstructions). Answered on page 12, lines 13-21 in the revised manuscript.

What have we learnt from this discussion and how do we respond to this criticism? (i) Error bars for $^{87}Sr/^{86}Sr$ data will be added to figures 3, 4, 5. (ii) The reference to the C7 segment will be removed from figure 8D, given that it is not discussed in the text. (iii) The paper certainly has benefitted from this discussion and we will add some text to the discussion chapter that deals with these complex issues. Error bars were added for $^{87}Sr/^{86}Sr$ ages in figures 3, 4, and 5. The reference to the C7 segment was removed from figure 9D (originally figure 8D).

However, our data point to an increase in seawater oxygenation at the end of C3 or at the base of C4. We agree that the upper part of the Kanfanar section is difficult to interpret due to a lack of automicrite that provides a reliable archive for the proxies applied here. Nevertheless, U-isotope ratios and Ce-anomalies indicate more oxygenated conditions of seawater upsection of the C3 segment. We feel that the U isotope data are the most reliable ones and given the long residence time of U in the global seawater, we would rather expect a global and not a regional signature. Moreover, this interpretation is supported by the $\delta^{13}$C signatures of well-preserved rudist calcite (Huck et al., 2010) recording values of 3-4 ‰. A sentence for the $\delta^{13}$C signatures of well-preserved rudist calcite (Huck et al., 2010) was added to the discussion on page 14, line 5 in the revised manuscript.

Concluding: In this study, we do not limit platform-top hypoxic events to the C3 segment and we will make this very clear (in case it was not clear in the previous version) in the revised version. Assuming the most severe seawater oxygen depletion at the end of C3, the platform-top is possibly also hypoxic in the C4 segment, but then shows slow recovery to more oxygenated conditions in the Kanfanar section. We would not assume that this pattern must be directly translated to that found in the Mljet section and the Husinec and Read data. These are "simply" very complicated patterns and our means to date and correlate are less than ideal.

Thank you for your very constructive comments,

Sincerely yours,

A. Hueter on behalf of the authors.

Dear Dr. Algeo,

Thank you very much for carefully reading our manuscript and your detailed and insightful comment. This discussion will significantly help to improve the quality of this study. Please find our step by step reply (AR) to the Referee comments (RC) below:

*RC#1: What is the water depth on the platform top and would the Great Bahama Bank represent a better example than the Late Pennsylvanian Midcontinent Sea of North America?*

AR#1: Due to the location on an elevated high at the northern edge of the Adriatic Carbonate Platform (ACP), we expect an overall shallow marine setting. This is also what the carbonate facies tells us. Water depths above 100 m are unlikely. Given the dimensions of the ACP, however, water depths of 100 and more meters are expected elsewhere. Strictly referring to the portion of the ACP we discuss here, yes, the setting would indeed be similar to the Great Bahama Bank. Overall, when comparing ACP and LPMS with mean water depths of ~50 m, the LPMS might be a better analogue due to its complexity, dimension, and wide range of palaeobathymetries. Summing up, it makes certainly sense to refer to the Bahamas AND the LPMS depending on the question in mind. Answered on page 13, lines 24-27 in the revised manuscript.

*RC#2: Flow of hypersaline water masses leaving the platform top and creating a mechanism to drive lateral advection or upwelling of open-ocean waters onto the platform top.*

AR#2: Assuming usual paleo-temperature reconstructions of the Cretaceous (Larson and Erba, 1999; Jenkyns, 2003), a greenhouse setting associated with strong evaporation on the platform-top is likely. This process would strengthen our hypothesis of upwelling hypoxic basinal water masses on the platform-top. Nevertheless, the impact of such a mechanism and its implications needs to be further investigated and will be discussed in a revised version of the manuscript. Answered and discussed on page 12, lines 32-36 in the revised manuscript.

*RC#3: Al concentration curve plotted next to redox sensitive trace elements (As, V and Mo).*

AR#3: Figures will be added in a revised version of the manuscript, showing the redox sensitive trace element values, normalized to Al. These figures provide information that Al hardly influences the redox sensitive trace elements and does not weaken their information concerning seawater oxygen levels. Furthermore, a figure will be added to demonstrate the Al content of every sample, following Ling et al. (2013), who established an Al maximum value of <0.35 % for a primary seawater signal. In our samples we observe values below 0.15 %, mostly around 0.05 %. An Al concentration curve (ppm) was plotted next to the redox sensitive trace elements in figure 4 in the revised manuscript.

*RC#4: Redox sensitive trace elements reflect platform-top conditions.*

AR#4: In general, the redox sensitive trace elements show the redox conditions on the platform top. Due to the increased uptake of these elements during basinal black shale deposition, their concentration will be reduced on the platform top, where they indirectly provide

information about the redox conditions in deeper areas. Answered on page 9, lines 10-13 in the revised manuscript.

*RC#5: Do REEs represent a hydrogenous source?*

AR#5: In order to avoid a detrital origin of the REE signal, all samples were carefully screened (thin sections and SEM analysis) for the presence of automicrite. The term "automicrite" was defined by Neuweiler and Reitner (1992) and refers to micrite that formed *in situ* by microbial activity inducing the nucleation and precipitation of mainly fine-grained Mg calcite crystals. Answered on page 6, lines 6-7 and page 8, lines 24-39.

It is here important to note that it has been previously documented that automicrite is an excellent archive of paleo-seawater chemical composition, notably for the REE pattern (Olivier and Boyet, 2006; Della Porta et al., 2015). A large majority of the REE pattern measured in automicrite in our study show a typical marine pattern, supporting the interpretation that the REE represent a hydrogenous source. These pattern will be shown in an updated version of our manuscript. Furthermore, figures of the Y/Ho ratio, Sc concentration and LREE/HREE ratio will be added, supporting hydrogenous source of the REEs. Answered on page 8, lines 26-32 and REE pattern were added in the new figure 7 in the revised manuscript.

In this reply, we provide a first raw version of the new figures in the attached material.

The Y/Ho ratio remains between 40 to 50 for most of the samples. Some samples show even higher ratios, but no sample is showing values below 35, indicating a terrigenous source. Answered on page 8, lines 32-33 and in figure 7 in the revised manuscript.

The LREE/HREE ratio is below 1.0 for all of the samples but we can observe an increase in the ratio during the complete section from values around 0.4 (section meter 12) to values close to 0.9 (section meter 27). Answered on page 8, lines 33-35 and in figure 7 in the revised manuscript.

The Al content is quite low and stable and mostly oscillates around 500 ppm (0.05%), significantly below the suggested maximum for a hydrogenous source of 0.35% (Ling et al., 2013). The same applies for Sc, where the highest measured concentration in our samples is close to 0.3 ppm, much lower than the maximum concentration (for seawater) of 2 ppm, as suggested by Ling et al. (2013). Answered on page 8, lines 35-39 and Al concentration curve was added in figure 4 in the revised manuscript.

*RC#6: U isotope ratios and Ce/Ce* record global signal to reducing conditions in the larger Tethys Ocean basin. Only the redox sensitive trace elements provide information about platform top and if they are controlled by clay content, there is no evidence for changes in redox conditions on the platform top.*

AR#6: The Al and Sc concentrations show that the concentrations of redox sensitive trace elements are not controlled by a change in the clay content.

We agree that uranium isotope ratios are likely to record a shift towards more reducing conditions in the Tethys Ocean basin. Due to its short residence time (compared to uranium) of a few hundred years, Ce/Ce* can still be used to reconstruct local palaeoredox conditions

(Sholkovitz and Schneider, 1991; Shields and Stille, 2001). Answered on page 8, lines 40-41 and on page 9, lines 1-4 in the revised manuscript.

*RC#7: Difficult to make a shallow watermass oxygen-depleted because of the fast exchange with the atmosphere.*

AR#7: Indeed, there is an exchange of oxygen with the atmosphere. But given the greenhouse climate combined with strong evaporation rates during the Early Aptian, ongoing carbonate production (due to the microencrusters mass occurrence) as well as the production and integration of $CO_2$ in the open ocean (Ontong Java; Jenkyns, 2003) the loss of oxygen could have been stronger than the absorption of oxygen from the atmosphere. We here refer to the kettle-effect described by Skelton and Gili (2012). Please also note that the present world knows > 300 oxygen depleted coasts, implying that yes, it is possible to make a shallow water mass oxygen depleted. This also depends on wave activity, as waves are major agents in actively transporting gasses into the ocean surface water. In a protected, low-energy setting and low wave activity (due to wave-seafloor interaction), these processes seem to work. Answered on page 12, lines 39-41 and on page 13, lines 1-7 in the revised manuscript.

*RC#8: What is the exact origin of automicrite?*

AR#8: Automicrite was defined by Neuweiler and Reitner (1992) and refers to micrite that formed *in situ* by microbial activity inducing the nucleation and precipitation of mainly fine-grained Mg calcite crystals. Answered on page 6, lines 6-7 and already in the first version of the manuscript, now page 6, lines 7-15 in the revised manuscript.

*RC#9: What are the petrographic or field criteria used to distinguish automicrite from detrital micrite?*

AR#9: We provide several text blocks in the paper that deal with this point. Here is the short version: In the field, automicrite is recognized as a fine-grained homogenous carbonate ooze with a characteristic weathering color that differs from the host detrital micrite. In this study, the occurrence of automicrite is often associated with the occurrence of the microencrusting organisms *L. aggregatum* and *B. irregularis*. Note, field observations need testing under the SEM!

A recrystallization of the material would lead to an increase of different grain sizes, and generally a shift away from a rather homogenous to an inhomogeneous grain size. We found no evidence for this during SEM analysis. We acknowledge the work of Algeo et al. (1992) and will add a paragraph on the diagenesis of automicrite in the discussion part of a revised version of the manuscript. Answered on page 4, lines 15-23 and page 6, lines 6-7 in the revised manuscript.

*RC#10: How is it possible to show changes in redox conditions and not in some covariant factor (such as water temperature) was the main biotic stressor?*

AR#10: What we learnt from recent international research initiatives such as BIOACID is the following: Most marine organisms tolerate one stressor to some degree. They, however, give

up rather quickly if for example high temperatures PLUS low pH act in parallel, to give one example. Here, we explicitly do not exclude high seawater temperatures and salinity etc. as additional stressors and we state this in the discussion. Obviously, in a fossil setting such as this one here, the fundamental question, which of the stressors (oxygen level, salinity, temperature, pH etc) was the main "killer" must remain to some degree unresolved. We have some circumstantial evidence though. For example, the high carbonate production rates are not in agreement with low seawater pH. We see a decline of oxygen sensitive organisms and a "bloom" of organisms that are (arguably) less sensitive to low oxygen levels etc.

The focus of this study is on oxygen depletion as driving mechanism for the biological change on the platform top. Arguments include the fact that biotic patterns are in parallel to patterns in seawater dissolved oxygen levels. Temperature as well as salinity etc. may have added to the overall "doom scenario" for rudist-coral communities. Answered on page 11, lines 32-35, page 13, lines 34-41 and on page 14, lines 1-2 in the revised manuscript.

Minor matters:

- Past events being described used present tense
  - Will be corrected in a revised version of the manuscript Several changes were made in the revised manuscript, including tense and spelling.
- What is an "out-of-balance" ecosystem?
  - An ecosystem that recorded a sudden change in dominant biota (a term commonly used by ecologists) Sentence was added on page 2, lines 33-34.
- What is the evidence for an increased sedimentation rate?
  - A change in the microencruster morphology (increase of vertical growth forms) Explanation was added on page 6, line 23.
- Some grammatical problems
  - Will be taken care of in the revised version Changed throughout the manuscript.

Thank you for your very constructive comments,

Sincerely yours,

A. Hueter on behalf of all co-authors.

Algeo, T.J., Wilkinson, B.H. and Lohmann, K.C. (1992): Meteoric-burial diagenesis of Middle Pennsylvanian limestones in the Orogrande Basin, New Mexico: Water/rock interactions and basin geothermics. *Journal of Sedimentary Petrology*, 62(4), https://doi.org/10.1306/D426797E-2B26-11D7-8648000102C1865D.

Algeo, T.J., Heckel, P.H., Maynard, J.B., Blakey, R.C. and Rowe, H. (2008): Modern and ancient epeiric seas and the super-estuarine circulation model of marine anoxia. In: Dynamics of Epeiric Seas, B.R. Pratt, and C. Holmden, eds. (*Geological Association of Canada Special Paper*), 8-38.

Della Porta, G., Webb, G.E. and McDonald, I. (2015): REE patterns of microbial carbonate and cements from Sinemurian (Lower Jurassic) siliceous sponge mounds (Djebel Bou Dahar, High Atlas, Morocco). *Chem. Geol.*, 400, 65-86, https://dx.doi.org/10.1016/j.chemgeo.2015.02.010.

Jenkyns, H.C. (2003): Evidence for rapid climate change in the Mesozoic-Palaeogene greenhouse world. *Phil. Trans. R. Soc. Lond. A*, 361, 1885-1916, https://doi.org/10.1098/rsta.2003.1240.

Larson, R.L. and Erba, E. (1999): Onset of the mid-Cretaceous greenhouse in the Barremian-Aptian: Igneous events and the biological, sedimentary, and geochemical responses. *Paleoceanography*, 14(6), 663-678.

Ling, H., Chen, X., Li, D., Wang, D., Shields-Zhou, G.A. and Zhu, M. (2013): Cerium anomaly variations in Ediacaran-earliest Cambrian carbonates from the Yangtze Gorges ares, South China: Implications for oxygenation of coeval shallow seawater. *Precambrian Research*, 225, 110-127, https://doi.org/10.1016/j.precamres.2011.10.011.

Neuweiler, F. and Reitner, J. (1992): Karbonatbänke mit *Lithocodium aggregatum* ELLIOTT / *Bacinella irregularis* RADOICIC. *Berliner geowiss. Abh.*, 3, 273-293.

Olivier, N. and Boyet, M. (2006): Rare earth and trace elements of microbialites in Upper Jurassic coral- and sponge-microbialite reefs. *Chem. Geol.*, 230, 105-123. https://doi.org/10.1016/j.chemgeo.2005.12.002.

Shields, G. and Stille, P. (2001): Diagenetic constraints on the use of cerium anomalies as palaeoseawater redox proxies: an isotopic and REE study of Cambrian phosphorites. *Chem. Geol.*, 175, 29-48. https://doi.org/10.1016/S0009-2541(00)00362-4.

Sholkovitz, E.R. and Schneider, D.L. (1991): Cerium redox cycles and rare earth elements in the Sargasso Sea. *Geochim. Cosmochim. Acta*, 55, 2737-2743, https://doi.org/10.1016/0016-7037(91)90440-G.

Skelton, P.W. and Gili, E. (2012): Rudists and carbonate platforms in the Aptian: a case study on biotic interactions with ocean chemistry and climate. *Sedimentology*, 59, 81-117, https://doi.org/10.1111/j.1365-3091.2011.01292.x.

[Figure]

[Figure]

**Fig. 1:** REE patterns of all samples used for calculating the Ce-anomaly values. The patterns indicate seawater origin and an enrichment of HREE compared to LREE. For terrigenous source, a bell-shaped pattern would be expected.

**Fig. 2:** The Y/Ho-ratios plotted against the profile meters (left side) and against ΣREE. All samples are ranging between a Y/Ho ratio of 36 and 70, with a cluster around a ratio of 45, clearly indicating a hydrogenous source of the measured signal.

[Figure]

**Fig. 3:** The concentration of Al plotted against the section meters. Al concentration remains stable throughout the complete measured interval. All values are significantly below the suggested maximum of 3500 ppm for seawater origin (Ling et al., 2013).

[Figure]

**Fig. 4:** Redox sensitive trace element concentrations normalized to Al concentrations. The trend of increasing trace element concentrations from section meters 8 to 10, followed by a decrease at section meter 13.5 is confirmed. An extremely low Al concentration value at section meter 21.2 is seen in all of the trace elements, but absent in the original Fig. 4, meaning that this Al value is not influencing the redox reconstruction for this part of the section. Since a single sample has such a low Al concentration, surrounded by samples with significantly higher Al concentrations, a measurement error must be assumed. However, an overall trend of Al having highly influenced the redox sensitive trace elements cannot be confirmed.

**Table 1:** Rare Earth Elements normalized to the Post Archaen Australian Shale (PAAS).

| Sample | La | Ce | Pr | Nd | Sm | Eu | Gd | Tb | Dy | Ho | Er | Tm | Yb | Lu |
|---|---|---|---|---|---|---|---|---|---|---|---|---|---|---|
| KAN 12.8 | 0.031 | 0.015 | 0.016 | 0.016 | 0.016 | 0.022 | 0.025 | 0.025 | 0.026 | 0.032 | 0.030 | 0.023 | 0.024 | 0.026 |
| KAN 13.2 | 0.040 | 0.013 | 0.019 | 0.021 | 0.022 | 0.031 | 0.032 | 0.034 | 0.035 | 0.042 | 0.041 | 0.031 | 0.032 | 0.035 |
| KAN 14.2 | 0.014 | 0.004 | 0.005 | 0.005 | 0.006 | 0.008 | 0.011 | 0.012 | 0.014 | 0.018 | 0.018 | 0.014 | 0.014 | 0.015 |
| KAN 15.4 | 0.017 | 0.005 | 0.007 | 0.007 | 0.008 | 0.012 | 0.013 | 0.015 | 0.017 | 0.022 | 0.023 | 0.017 | 0.017 | 0.019 |
| KAN 16.3 | 0.033 | 0.010 | 0.016 | 0.017 | 0.018 | 0.026 | 0.027 | 0.028 | 0.031 | 0.039 | 0.038 | 0.028 | 0.029 | 0.033 |
| KAN 16.6 | 0.032 | 0.012 | 0.016 | 0.017 | 0.018 | 0.025 | 0.026 | 0.028 | 0.030 | 0.037 | 0.037 | 0.028 | 0.030 | 0.034 |
| KAN 16.85 | 0.029 | 0.011 | 0.019 | 0.020 | 0.022 | 0.031 | 0.029 | 0.029 | 0.030 | 0.034 | 0.032 | 0.023 | 0.025 | 0.027 |
| KAN 20.8 | 0.023 | 0.012 | 0.016 | 0.018 | 0.019 | 0.027 | 0.028 | 0.027 | 0.027 | 0.030 | 0.028 | 0.020 | 0.022 | 0.023 |
| KAN 21.2 | 0.017 | 0.011 | 0.012 | 0.013 | 0.013 | 0.019 | 0.021 | 0.020 | 0.020 | 0.023 | 0.021 | 0.015 | 0.016 | 0.018 |
| KAN 21.85 | 0.016 | 0.008 | 0.010 | 0.011 | 0.011 | 0.017 | 0.020 | 0.019 | 0.020 | 0.023 | 0.020 | 0.015 | 0.016 | 0.016 |
| KAN 22.4 | 0.020 | 0.013 | 0.014 | 0.015 | 0.016 | 0.023 | 0.024 | 0.023 | 0.023 | 0.025 | 0.022 | 0.016 | 0.018 | 0.019 |
| KAN 22.7 | 0.020 | 0.016 | 0.017 | 0.017 | 0.018 | 0.025 | 0.022 | 0.022 | 0.022 | 0.024 | 0.022 | 0.017 | 0.019 | 0.021 |
| KAN 24.8 | 0.023 | 0.020 | 0.020 | 0.020 | 0.020 | 0.027 | 0.022 | 0.022 | 0.021 | 0.023 | 0.023 | 0.018 | 0.020 | 0.022 |
| KAN 25.2 | 0.018 | 0.017 | 0.017 | 0.018 | 0.020 | 0.026 | 0.020 | 0.020 | 0.019 | 0.019 | 0.017 | 0.013 | 0.014 | 0.015 |
| KAN 26.4 | 0.029 | 0.016 | 0.019 | 0.021 | 0.024 | 0.031 | 0.027 | 0.027 | 0.026 | 0.028 | 0.025 | 0.018 | 0.019 | 0.020 |
| KAN 26.6 | 0.025 | 0.014 | 0.014 | 0.015 | 0.017 | 0.022 | 0.019 | 0.018 | 0.018 | 0.018 | 0.017 | 0.012 | 0.013 | 0.013 |
| KAN 27.0 | 0.030 | 0.012 | 0.013 | 0.014 | 0.016 | 0.021 | 0.021 | 0.019 | 0.018 | 0.019 | 0.017 | 0.011 | 0.010 | 0.011 |
| KAN 27.6 | 0.048 | 0.015 | 0.015 | 0.017 | 0.019 | 0.024 | 0.030 | 0.024 | 0.024 | 0.027 | 0.023 | 0.015 | 0.014 | 0.014 |

Dear Dr. Weissert,

Thank you very much for carefully reading our manuscript and your detailed and insightful comments.

*RC#1: Stratigraphic interpretation of the Kanfanar section*

AR#1: As you already noticed, the biostratigraphy of the Kanfanar section is based on the occurrence of benthonic foraminifera (*Palorbitolina lenticularis*) as established in a previous study by Huck et al. (2010). We are aware of the difficulties regarding the stratigraphy of shallow-water carbonates and open for any further interpretations. The carbon-isotope pattern established for the Monte Faito section by Amodio and Weissert (2017) reveals many similarities with the Kanfanar carbon-isotope record. Having a closer look at the $\delta^{13}C$ pattern of the Monte Faito location, the positive peak of +4 ‰ (section meter 86) fits nicely with the +4 ‰ positive excursion (around section meter 30) in the Kanfanar section (rudist data). If this assumption is correct, the mass occurrence of microencrusting organisms (*Lithocodium aggregatum* and *Bacinella irregularis*) would not represent the Selli Level Equivalent, the latter being masked by a regional exposure surface higher up in the Kanfanar section (at section meter 32.5). This would lead to the interpretation of a pre-OAE 1a age for the studied microencruster interval in the Kanfanar section. Answered on page 4, lines 5-9; page 7, lines 30-36 and page 8, lines 1-20 in the revised manuscript.

On the other hand, Huck et al. (2010) compared the $\delta^{13}C$ pattern of the Kanfanar section with the patterns of a composite section in Oman and the La Bedoule Section in France, presenting many similarities as well. These similar patterns share the strong positive peak in $\delta^{13}C$ of +4 ‰ (rudist shells in the Kanfanar section) and would rather support the current interpretation of the stratigraphic framework, additionally encouraged by Sr ages (Oman and Croatia). See page 8, lines 10-15 in the revised manuscript.

Nevertheless, we are thankful for this helpful interpretation to this apparently very complex topic and will not exclude this possibility, due to the already known occurrences of *L. aggregatum* and *B. irregularis* before and after OAE 1a (e.g. Portugal and Oman). As you already stated, the connection of hypoxia or even anoxia and the massive occurrence of microencrusting organisms would not be affected by this new interpretation. Anyway, the biostratigraphy and the possible connection between OAE 1a and microencruster occurrences is very important for the understanding of these complex cause and effect patterns. Therefore, we will discuss this possibility in detail in a revised version of this manuscript. See page 8, lines 16-20 in the revised version.

*RC#2: Maximum of oxygen depletion (as seen in uranium isotope ratios and Ce-anomaly values) coincides with the maximum abundance of microencrusting organisms at the top of chemostratigraphic segment C3.*

AR#2: In your comment, you are referring to text that states that the maximum oxygen depletion (as seen in the geochemical proxies) coincides with the maximum abundance of microencrusting organisms at the top of chemostratigraphic segment C3. We agree that this statement is somehow misleading, as figure 3 shows the maximum abundance at section meter 17 and 22. Indeed, the abundance is already decreasing at the top of C3. The text should probably better read: At the top of the C3 segment, uranium isotope ratios and cerium anomalies indicate a maximum oxygen depletion, but the abundance of microencrusters is already decreasing. There are two possible explanations for this: (1) Regarding the complex morphology of L. aggregatum and B. irregularis (e.g. Rameil et al., 2010), the two-dimensional perspective in thin sections may induce some degree of data bias, i.e. the thin section area covered by microencrusters strongly depends on the plane of observation. Moreover, the abundance and the grown morphology may change due to changing environmental parameters. We are aware of this problem and took great care to compare observations from the field and thin sections observation to reduce any potential error to a minimum. Despite all our care, we cannot exclude, however, that some of the second order patterns are not, to some degree, influenced by these issues. In short, this is a complex, non-perfect, geological system and any sampling strategy may induce this type of problems. We are confident that the first order patterns are valid and can be replicated though. (2) The more likely interpretation is: The oxygen depletion is reaching its maximum at the top of C3 and becomes critical for even the hardiest organisms, exceeding their tolerance limits and hence they decline. We will change the relevant statements in the manuscript and discuss the possible explanations for the decrease in microencruster abundance despite persistent oxygen deficiency. Answered on page 9, lines 32-41 and page 10, lines 1-2 in the revised manuscript.

As a second important point of your second comment refers to the discussion of our geochemical redox proxies. We agree that uranium isotope ratios will preserve a global signal due to the long residence time of uranium in the global ocean (450 kyr). In contrast, cerium, with a residence time of a few hundred years, as well as the redox sensitive trace elements, will display a local or regional signal (Sholkovitz and Schneider, 1991; Shields and Stille, 2001; Bodin et al., 2007), supported by the long mixing time of the oceans (~1000-2000 years). For this reason, we will add some text in the discussion that deals with the temporal level on which the redox proxies operate. Answered on page 8, lines 40-41 and on page 9, lines 1-4 in the revised manuscript.

Minor comments: (1) P.2, line 23: We will add Wissler et al. (2003) as a further reference. Reference was added on page 1, line 24. (2) P.7, line 15: It should be "(at 26 m)" instead of "(2at 6 m)". Will be changed in a revised version. Was changed on page 7, line 25. (3) P.8, line 38: We will add some new references and discuss the question of upwelling or downwelling. Martin et al. (2012) was added and upwelling/downwelling was discussed on page 10, lines 33-38 in the revised version. (4) P.11, line 23: We will discuss the possible impact of Barremian black shale deposition on the studied interval. Sentence was added on page 14, lines 18-20 in the revised version. (5) P.11, line 32: We will cite Méhay et al. (2009) to provide detailed information about the changes in pCO2 at the base of OAE 1a. Méhay et al. (2009) was added on page 14, line 30.

Thank you for you very constructive comments.

Sincerely yours,

A. Hueter on behalf of the co-authors.